

# Global scenarios of anthropogenic mercury emissions

Flora Maria Brocza[1,2], Peter Rafaj[1], Robert Sander[1], Fabian Wagner[1], Jenny Marie Jones[2]

[1]International Institute for Applied Systems Sciences, Schlossplatz 1, 2361 Laxenburg, Austria
[2]School of Chemical and Process Engineering, University of Leeds, LS2 9J, United Kingdom

*Correspondence to*: Flora M. Brocza (brocza@iiasa.ac.at)

**Abstract.** Anthropogenic mercury (Hg) emissions to the atmosphere are a long-lived hazard to human and environmental health. The UN Minamata Convention on Mercury is seeking to lower anthropogenic mercury emissions through a mix of policies from banning Hg uses and trade, to reducing unintentional Hg releases from different activities. In addition to independent Hg policy, greenhouse gas, particulate matter (PM) and $SO_2$ reduction policies may also lower Hg emissions as a

co-benefit. This study uses the Greenhouse Gas – Air Pollution Interactions and Synergies (GAINS) model to examine the effect of different clean air and climate policy on future global Hg emissions. The Baseline scenario assumes current energy use and Hg emissions, as well as current legislation for clean air, mercury and climate policy. In addition, we explore the impact of the Minamata Convention, co-benefits of climate policies and of stringent air pollution policies, as well as a maximum feasible reduction scenario for Hg (Hg-MFR). Hg emission projections until 2050 show noticeable reductions in

combustion sectors for all scenarios, due to a decrease in global fossil fuels and traditional biomass use, leading to emission reductions of 33% (Baseline) up to 90% when combining stringent climate and Hg-MFR. Cement and non-ferrous metal emissions increase in all activity scenarios with current air pollution policy, but can be reduced by up to 72% and 46% respectively in 2050 with stringent Hg-specific measures. Other emissions (including waste) are a large source of uncertainty in this study, and projections range between a 22% increase and 54% decrease in 2050 depending on both climate and clean

air policy. The largest absolute reduction potential for Hg abatement, but also the largest uncertainties of absolute emissions lie in the in small-scale and artisanal gold production, where Hg-specific abatement measures could eliminate annual Hg emissions in the range of 601-1371 t (95% confidence interval). 90% of the Hg emissions in GAINS are covered by the Minamata Convention. Overall, the findings emphasize the necessity of implementing targeted Hg control policies in addition to stringent climate, PM and $SO_2$ policies to achieve significant reductions in Hg emissions.




**Graphical Abstract.**

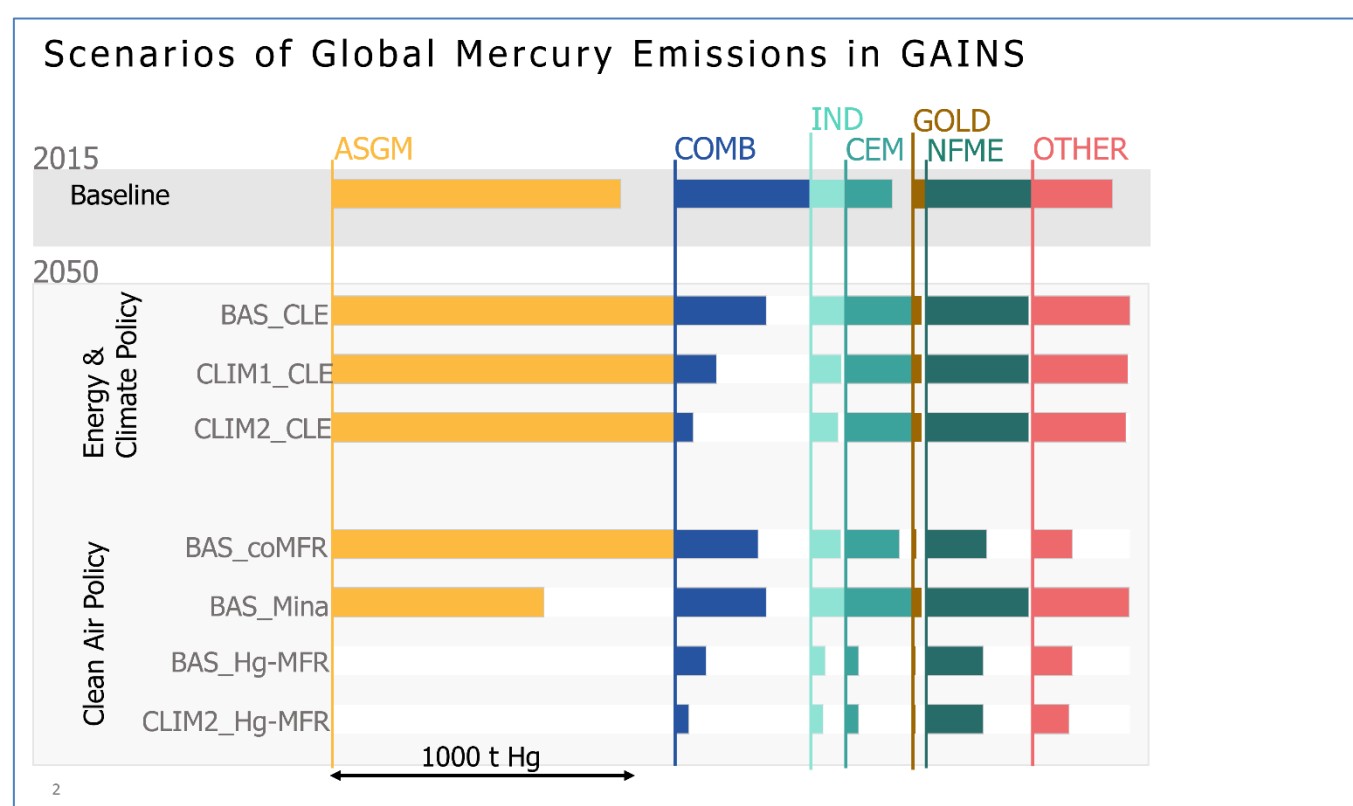

**Abbreviations**

| | | |
|---|---|---|
| 30 | ASGM | Artisanal and small-scale gold mining |
| | BAS | Baseline scenario |
| | BAT | Best available technology |
| | BEP | Best environmental practice |
| | CLE | Current legislation |
| 35 | CLIM1 | Climate policy scenario |
| | CLIM2 | Net-zero scenario |
| | co-MFR | co-benefits for Hg from PM, $SO_2$ MFR |
| | EC | Emission control |
| | EF | Emission factor |
| 40 | EU-IED | Industrial Emissions Directive of the European Union |
| | GDP | Gross domestic product |



| | | |
|---|---|---|
| | GMA'18 | Global Mercury Assessment ((AMAP/UNEP 2019)) |
| | Hg-MFR | Maximum feasible reduction for Hg |
| | MEX | Market exchange rate |
| 45 | MFR | Maximum feasible reduction |
| | MCM | Minamata Convention on Mercury |
| | Mina | Minamata policy scenario |
| | NAP | National Action Plan |
| | NFME | Non-ferrous metals |
| 50 | NOC | No Control Scenario |
| | PM | Particulate matter |
| | PPP | Purchasing power parity |
| | POP | Population |
| | SRES | IPCC Special Report on Emission Scenarios |
| 55 | UEF | Unabated emission factors |
| | VCM | Vinyl chloride monomer production |
| | WEO | World Energy Outlook |
| | WHO | World Health Organization |

# 1 Introduction

Mercury (Hg) is one of the top ten chemicals of major public health concern designated by the World Health Organization (WHO). The metal's unique volatility and (redox-)reactivity at ambient conditions facilitate frequent Hg-species changes, leading to long-range atmospheric transport, subsequent deposition and re-emission of the metal and its derivative compounds, as well as bioaccumulation of the most toxic Hg species, methyl mercury, in the aquatic food chain. The extent of the pollution and health problems caused by atmospheric Hg emissions has been known for two decades (e.g. UNEP Chemicals 2002).

Cumulative anthropogenic emissions have increased the Hg content in the atmosphere by 450% above natural levels (AMAP/UNEP 2019). The time for mercury to return to a permanent sink such as deep ocean sediments has been estimated as up to 3000 years (Selin 2009), demonstrating that Hg pollution will continue to pose a serious environmental threat for years to come, but also highlighting that today's action will have a long-lasting effect to reduce levels of environmental Hg (Angot et al. 2018). From a health perspective, it has been estimated that accumulated health effects of Hg pollution will cost $19

trillion globally between 2010 and 2050 (2020 dollars; Zhang et al. 2021), further demonstrating the importance of fast action. To break the cycle of emissions, re-emissions and heightening pollution, the Minamata Convention on Mercury (MCM) has been adopted in 2013. It entered into force in 2017 and is presently ratified by 147 countries (UNEP 2013). The first international health and environment treaty on hazardous substances in almost a decade, it recognizes that Hg emissions must



be tackled urgently at the global level. The MCM aims to reduce releases of "mercury and mercury compounds" by targeting

them at different levels of the release cycle, such as trade, use in production, use in products, emission sources, and wastes. Mercury releases to the atmosphere and environment are on one hand addressed by technical solutions, such as best available technology / best environmental practice (BAT/BEP) recommendations for Hg handling, industrial emissions or waste storage. On the other hand, they require political and regulatory action, such as bans on mercury trade, specific products, and small scale or traditional (artisanal) gold mining practices, demonstrating a "life-cycle approach" to limiting Hg emissions (e.g. Selin

2014; Giang et al. 2015). Despite these efforts, global anthropogenic emissions of Hg were estimated to have risen by 20% by 2015 compared to "pre-Minamata" 2010 levels. Small emission decreases in North America and the EU were offset by a mix of increased economic activity, as well as the production, use and disposal of mercury-containing products (AMAP/UNEP 2019; Pacyna et al. 2016). For a better understanding of future Hg levels in the atmosphere, scenarios of future anthropogenic Hg releases are needed. Such scenarios need to consider the wide range of Hg emission sources, their emission intensity, as

well as their drivers.

Where Hg is emitted in energy-intensive sectors, such as from the combustion of fossil fuels or different industries, future emissions strongly depend on the assumptions on future energy demand and the decarbonization of those sectors. Emission trends from other sectors, such as waste generation, are derived from macroeconomic factors and population growth. Other activities, such as artisanal and small-scale gold production, are specific to mercury pollution and untouched by other air

quality and climate policy. Emission intensity is always specific to the emission source, its geographic location and the application of control measures which lower the amount of Hg released into the atmosphere. Such measures include policies or technologies targeted directly at Hg. In addition, stringent clean air policy targeted at reducing particulate matter (PM), $SO_2$ and $NO_x$ is well known to lower Hg emissions, as the applied pollution control technologies interact with the mercury present in the flue gas streams and are able to retain it (e.g. Granite et al. 2000, Pavlish et al. 2003). Scenario analysis is a powerful

tool to quantify future pathways of anthropogenic mercury emissions and to understand interdependencies of various mitigation factors.

Only a small number of studies have produced global scenarios of speciated future mercury emissions; Streets et al. 2009 created a Hg inventory spanning different combustion and industry sectors as well as artisanal and small-scale gold mining (ASGM)[1]. These emissions were projected to 2050 based on four climate scenarios from the IPCC Special Report on Emission

Scenarios (SRES). The SRES is also used as a source of different energy scenarios for Hg projections by Lei et al. 2014. Rafaj et al. present one baseline and one climate scenario based on the World Energy Outlook 2012 and cover some Hg-specific sectors such as gold and caustic soda production; Pacyna et al. 2010, project Hg emissions up to 2020 based on different scenarios focusing on Hg-specific policies. Pacyna et al. 2016, projected Hg emissions up to 2035 based on the GMA'13 inventory and own projections, looking at a mix of scenarios including current legislation, maximum feasible reduction and a

---

[1] The Minamata Convention further defines ASGM as '*gold mining conducted by individual miners or small enterprises with limited capital investment and production*' (Minamata Convention on Mercury, 2013)





450 ppm $CO_2$ climate scenario. Additionally, several regional studies are available for China (Giang et al. 2015, Zhao et al. 2015, Ancora et al. 2016, Wu et al. 2018a,b, Mulvaney et al. 2020), India (Chakraborty et al. 2013, Giang et al. 2015) and Europe (Pacyna et al. 2006, Glodek et al. 2010 (Poland), Rafaj et al. 2014). The base years for the global scenario studies lie between 2000 and 2010 and the projection years are between 2020 and 2050. The most recent Global Mercury Assessment 2018 (GMA'18) was published for 2015 and includes significant data quality improvements compared to the 2010 inventory,

also including quantification of more emission sources. Similarly, our outlook on energy and climate scenarios has significantly changed since the COVID-19 pandemic and recent geopolitical developments. Up-to-date scenarios will be needed to understand future mercury emissions and the effectiveness of clean air and climate policy on curbing them.

IIASA's Greenhouse Gas and Air Pollution Interactions and Synergies (GAINS) model is uniquely suited for the creation of

global mercury emission scenarios (Amann et al. 2011). Originally built to inform policy questions regarding acid rain and particulate matter, its database was extended to Hg in 2013 (Rafaj et al. 2013; Rafaj et al. 2014). In GAINS, sector- and region-specific control strategies represent the pollution control measures and policies which are in place. Developments and policy in air pollution control, greenhouse gas reduction, as well as co-benefits from PM and $SO_2$ abatement and changes in the energy and industrial sectors are represented for each of the 182 GAINS regions. This study presents a recent update on the

methodology of accounting for co-benefit control of Hg from PM and $SO_2$ in the GAINS model. Dedicated mercury control options were also updated. We demonstrate the results of scenario analysis using three different energy/climate pathways combined with four different scenarios of mercury control measures, including mercury control options consistent with current legislation and Minamata commitments, as well as a maximum feasible reduction scenario. Scenarios were designed to identify the impact of climate policy, co-benefits from air pollution control policy, and dedicated Hg measures and are presented on

the level of 7 world regions and 8 exemplary sub-regions.

## 2 Modelling Framework

### 2.1 The GAINS model

The GAINS model quantifies emissions to the atmosphere, costs and (health) impacts of different strategies to reduce different air pollutants and greenhouse gases (Amann et al. 2011). GAINS computes Hg emissions on a global level up to 2050 in 5-

year time steps with a resolution of 182 regions. A region represents either a country, a group of neighboring countries or sub-national regions. Current and future emissions of mercury ($E^{Hg}$) are computed via equation (1) for each mercury species (f) from activity data (A) of different pollution sources - activity combinations (p,s) and uncontrolled emission factors (e), which are lowered by taking into account the removal efficiencies (r) of different emission control technologies and other measures (t) and their application rates (x) in a specific sector:



$$E^{Hg} = \sum_f E_{p,s,t} = \sum_{f,p,s,t} A_{p,s}\, e_{p,s}(1 - r_{f,t})x_{p,s,t}$$


(1)

An illustration of this is provided by Figure 1. In total, 13 fuel types used in 52 combustion sectors and 22 non-combustion emission sources are covered in GAINS, as summarized in the appendix (Tables S1, S2)[2]. For the purpose of this study, GAINS sector-activity combinations have been grouped into the main emission sources for Hg, described in Table 1. Similarly, the

182 GAINS regions are grouped into 7 main regions (Africa, Asia Pacific, Central and South America, Europe, Eurasia, Middle East, North America) in accordance with the IEA World Energy Outlook 2022. Additionally, Brazil, China, the European Union, India, Japan, Russia, Southeast Asia and the USA are computed separately as sub-regional case studies (see Table S10 in the appendix).

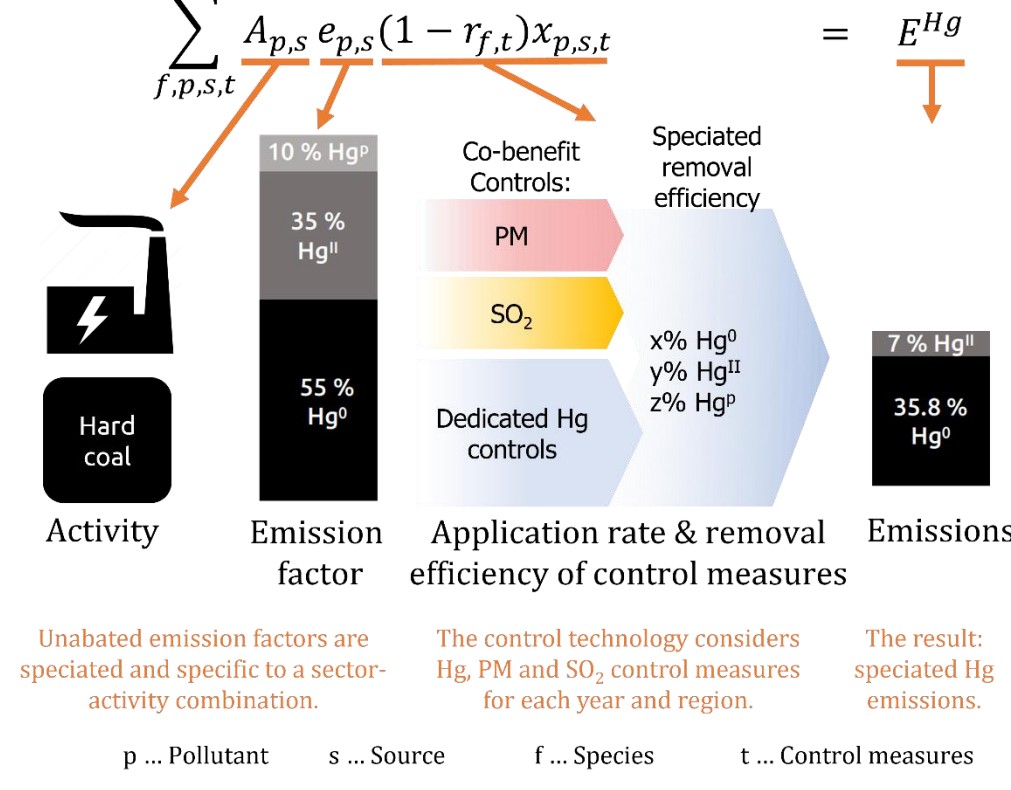

**Figure 1: Schematic of control technology application in GAINS.**

---

[2] The full list of activities, sectors and technologies in GAINS can be found in the GAINS glossary: https://gains.iiasa.ac.at/gains/GOD/abbreviations.info



## 2.2 Emission Factors

Uncontrolled emission factors (UEFs) are derived from literature sources and are specific to each sector-activity combination (see Table 1). Emission factors for hard coal and brown coal combustion remain unchanged from previous versions of GAINS (Rafaj et al. 2013; Rafaj et al. 2014). Factors for the production of cement, non-ferrous metals, aluminum, iron and steel, gold,

and caustic soda were updated in accordance with the Global Mercury Assessment 2018 ('GMA'18', AMAP/UNEP 2019). The non-ferrous metals (NFME) sector in GAINS includes emissions from copper, lead, nickel and zinc. Metal-specific emission factors (either country-specific or generic, depending on data availability) were weighted with the share of the relevant metals of the total activity for a particular GAINS region, based on USGS Mineral Yearbook production data (Klochko 2021; Flanagan 2022; Tolcin 2022). The shares were calculated for 2015 and assumed to be static, so this composite emission

factor was subsequently applied to all years for a particular region. Total gold production was similarly based on USGS Mineral Yearbook data. This data was split into country-level shares of large-scale (sector 'GOLD') and artisanal/small-scale gold mining (sector 'ASGM'), based on data from the World Gold Council as presented in GMA'18. Hg mining and vinyl chloride monomer production are not specified explicitly in the model and no control measures can be applied to them directly, but their emissions are included in the aggregate category "Other Hg emissions" on a region-by-region basis and their activities

reflect projected sectoral emissions (George 2021). In the waste sector, Hg emissions were derived from the GMA'18 and attributed evenly to industrial, rural and urban waste categories.

Information on the average speciation of emissions from each source are also implemented, dividing total unabated emissions into shares of $Hg^0$, $Hg^{II}$ and particulate Hg ($Hg_p$). Due to a lack of regional data, speciation data was implemented for each sector-activity combination on a global level. The values represent the best available literature data and modelled speciation

from the iPOG tool (Niksa Energy Associates LLC 2011) at the time of writing, but care has to be taken in their interpretation, as they are associated with large uncertainty. Table 1 summarizes the ranges of unabated emission factors used in the GAINS model for aggregated Hg-relevant sectors, as well as the $Hg^0$ / $Hg^{II}$ / $Hg_p$ of uncontrolled stack emissions.

**Table 1: Ranges of unabated emission factors (UEF) and speciation in GAINS on the global scale. UEFs vary on a regional scale, as**
**well as due to different, aggregated sector-fuel combinations. Bold categories represent the sector aggregation level that are plotted in the results figures (Figs. 3, 4, 5, S1).**

| Sector | Abbreviation | Emission factor | Speciation - inlet | Sources |
|---|---|---|---|---|
| | (as in Fig. 7) | min - max (unit) | $Hg^0$ / $Hg^{II}$ / $Hg_p$ | |
| *COMBUSTION - by sector* | | | | |
| Combustion in power plants | COMB_POWER | 0.0001 - 0.0477 (t/PJ) | 50-60 / 30-40 / 10 | Rafaj et al. 2013 |
| Industrial combustion | COMB_IND | 0.0001 - 0.063 (t/PJ) | 20-60 / 30-60 / 10-20 | Rafaj et al. 2013 |





| | | | | |
|---|---|---|---|---|
| Other combustion (Residential, service, conversion sectors) | COMB_OTHER | 0.0001 - 0.0477 (t/PJ) | 20-60 / 30-60 / 10-20 | Rafaj et al. 2013 |
| *COMBUSTION - by fuel* | | | | |
| All coals | | 0.0005 - 0.0477 (t/PJ) | 50-60 / 40-60 / 10-20 | Rafaj et al. 2013 |
| Gasoline | | 0.0001 (t/PJ) | 50 / 40 / 10 | Rafaj et al. 2013 |
| Liquid fuels | | 0.0001 - 0.0005 (t/PJ) | 50 / 40 / 10 | Rafaj et al. 2013, GMA'18 |
| Biomass | | 0.001 (t/PJ) | 50 / 40 / 10 | Rafaj et al. 2013, GMA'18 |
| Waste | | 0.063 (t/PJ) | 20 / 60 / 20 | Own estimate, derived from GMA'18 |
| INDUSTRY | | | | |
| Non-ferrous metals (Cu, Zn, Pb, Al) | NFME | 0.0002 - 117.84 (g/t) | 80 / 15 / 5 | GMA'18 |
| Large-scale gold | GOLD | 12000 – 55000 (g/t) | 80 / 15 / 5 | GMA'18 |
| Artisanal and small-scale gold | ASGM | 975000 – 1500000 (g/t) | 100 / 0 / 0 | GMA'18 |
| Cement production | CEM | 0.022 - 0.124 (g/t) | 80 / 15 / 5 | GMA'18 |
| Other industrial processes | IND_PROC | 0.00025 – 20 (g/t) | 70-80 / 15-30 / 0-5 | |
| Iron and steel production | | 0.0061 - 0.41475 (g/t) | 80 / 15 / 5 | Wang et al. (2016), GMA'18 |
| Oil Refining | | 0.0003 - 0.0166 (g/t) | 80 / 15 / 5 | GMA'18 |
| Caustic Soda Production | | 2.5 – 20 (g/t) | 70 / 30 / 0 | GMA'18 |
| OTHERS | | | | |
| Cremation | OTHER | 2 - 2.5 (g/Million) | 80 / 15 / 5 | Rafaj et al. (2013) |
| Waste | OTHER | 0.0315 (g/t) | 96 / 4 / 0 | GMA'18 |
| VCM production, Hg mining | OTHER | 1 (t Hg/year) | 100 / 0 / 0 | GMA'18 |
| Transport | OTHER | 0.0001 - 0.063 (t/PJ) | 20-60 / 30-60 / 10-20 | Own estimate, derived from GMA'18 |

Notes: GMA'18 … AMAP/UNEP (2019)





## 2.3 Control Technologies

### 2.3.1 Mercury control measures

A review of Hg control technologies and measures has been conducted. Relevant technologies have been implemented into
the GAINS model in addition to previously available co-benefit controls from PM and SO₂ abatement. Such new controls
include: the option of low-mercury or halogen-treated coal (LHGCO); sorbent injection (such as activated carbon) with or
without an additional baghouse filter (SINJ); acid plants for the non-ferrous metal and gold sectors (PR_AP); and stationary
sorbent modules (SPC), which represent the possibility of removing Hg not only from the atmosphere but bringing it into a
permanent sink such as a controlled hazardous waste landfill, rather than re-directing emissions into other environmental
releases. Removal efficiencies and Hg speciation of the control technologies operating on Hg are summarized for each emission
source category in the appendix (Tables S3-S6).

### 2.3.2 Quantification of co-benefits for mercury from particulate matter and SO₂ control

GAINS has been used extensively to inform policies on the reduction of particulate matter (PM), SO₂ and NOₓ, whose
abatement is known to strongly influence Hg emissions and their speciation. To compute impacts of traditional air pollution
control devices on total reduction of mercury in the GAINS model, current and projected control strategies of PM and SO₂ are
considered in addition to Hg-specific control measures[3]. The concept of 'overlapping control measures' has already been
introduced in an earlier publication (Rafaj et al. 2013). Where Hg-relevant PM and/or SO₂ co-exist in a region, sector and year,
their compounded impact Hg emissions is considered, increasing Hg removal efficiency for the relevant portion of installations.
Where appropriate, combinations of Hg-specific measures and PM/SO₂ measures also lead to increased Hg removal efficiency.
The relevant technology combinations are listed in Tables S4-S6 and have sector- and activity- specific removal efficiencies.
Relevant technology combinations are in the power and industry sectors between different particle filters and flue gas
desulphurization, and acid plants in industrial processes. Lastly, there has been a significant update of the representation
emissions from waste in GAINS in the past years (Gómez-Sanabria et al. 2022). Control measures in the GAINS waste sector
are multi-pollutant controls representing different types of landfill and other waste management options, not all of which can
be linked to reduced atmospheric Hg emissions. In this modelling work, the following three measures have been associated
with Hg removal efficiencies, as Hg reduction can be expected from literature review: waste incineration with energy recovery
and pollution controls (emissions accounted for in the power sector), landfill compression, landfill covering. Details can be
found in the appendix, Table S6.

---

[3] Co-benefits for NOx were not implemented as NOx control, when combined with PM and SO₂ control, has been reported to
only bring Hg removal efficiency improvements of a few percent lower than the standard deviation of the constructed
technology categories used in this study (see e.g., Li et al. 2020 (SI, Table S10), or the iPOG tool at reference conditions (Niksa
Energy Associates LLC, 2011).





### 2.3.3 Effect of control measures on Hg speciation

Mercury speciation is also altered through the application of control measures. The speciated removal efficiency of the controls operate on the speciated no-control emissions. The resulting total Hg emissions are then split again into $Hg^0$ / $Hg^{II}$ / $Hg_p$, based on reported stack speciated emissions of the technology from literature review. Tables of all applied control technologies, their speciated removal efficiencies and resulting emission speciation can be found in the appendix (Tables S3-S6).

### 3 Activity Projections

GAINS uses exogenous projections of anthropogenic activities and energy use to estimate future Hg emissions. For this study, three scenarios of energy and industrial production until 2050 were implemented in GAINS, based on trends reported by the World Energy Outlook 2022 (IEA 2022). The three scenarios share assumptions on macroeconomic drivers (GDP growth, GDP per capita and annual population growth). They differ in their assumption on the stringency of climate policy and already include first effects of the Russian war in Ukraine. While the total, global energy demand either rises or stays similar to 2015 levels until 2050 in all three scenarios, there are differences in the energy sources which meet this demand (see Fig. 2). The Baseline (BAS) scenario represents developments in energy leading to a 2.5C average global temperature rise by 2100 and is characterized by plateauing emissions in at 37 Gt and a reduction to 32 Gt energy-related $CO_2$ emissions in 2050. The demand growth is mostly met by renewable sources and the share of fossil fuels falls to 60% in 2050. The Climate Policy (CLIM1) scenario is consistent with a global average temperature rise of 1.7 C by 2100. Demand for all fossil fuels already declines by 2030, leading to a decrease in $CO_2$ emissions to 12 Gt by 2050. The Net Zero (CLIM2) scenario is the only scenario leading to <1.5 C temperature rise until 2100 (IEA 2022). The scenarios are summarized in Table 2.





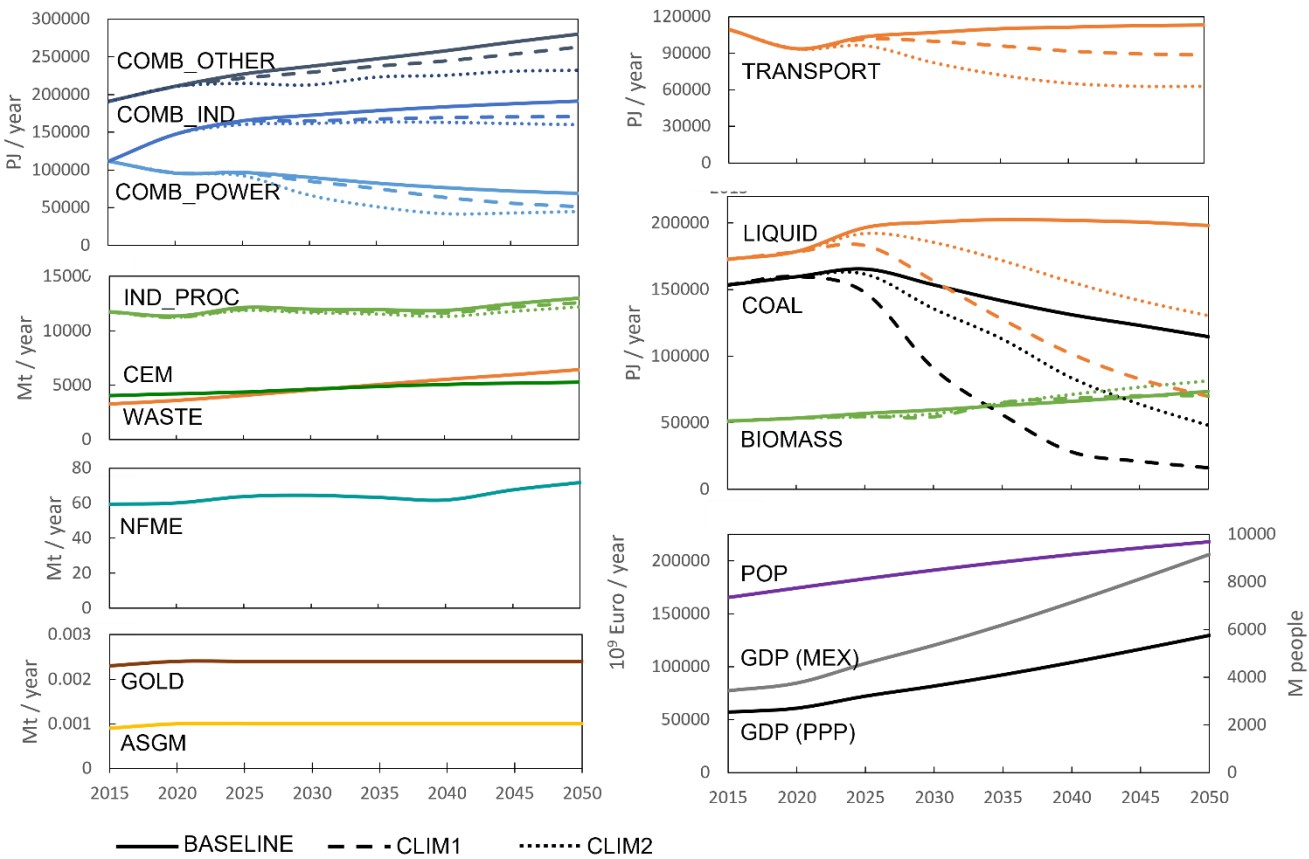

**Figure 2: Key activity data for the BASELINE, CLIM1 and CLIM2 scenarios. COMB_POWER ... Power plants, COMB_IND ...**
**Combustion in industry, COMB_OTHER ... Other combustion (residential, commercial, conversion losses), IND_PROC ... Industrial Processes, CEM ... Cement, NFME ... Non-ferrous metal production, ASGM ... Artisanal and Small-scale gold mining, POP ... Population, GDP ... Gross domestic product, MEX ... Market exchange rate, PPP ... Purchasing power parity. Projections of fuel use (COMB and different fuels), transport, IND_PROC, CEM, NFME, POP & GDP from the World Energy Outlook 2022 (IEA 2022). Waste data from Gomez-Sanabria et al. 2022. PGOLD and ASGM ... this study (see Section 3.)**


In addition, activities of Hg-specific sectors[4] have been modelled in GAINS as follows. Gold production volumes have been cross-checked between the USGS Mineral Yearbook data (Sheaffer 2022) and reports on ASGM from the World Gold Council via GMA'18, based on which shares between small-scale and large-scale gold production were calculated for 2020. Where the ASGM amounts in a specific country were larger than reported total gold production by the USGS, this higher number was 230   taken and the share of ASGM was assumed to be 1. Shares between ASGM and large-scale gold production were then taken as fixed for all past and projection years. Gold production and ASGM activity have been projected into the future following

---

[4] 'Hg-specific sector' means: a sector which, in the GAINS model, is only associated with Hg emissions (e.g. no PM, $SO_2$ or other emissions)



the conservative assumption that levels will stay the same to 2020 levels, only to be modified by changing Hg policy through the control strategy (e.g., a ban on ASGM).

Caustic soda production from chlor-alkali plants using mercury cells has been adopted from the Rafaj et al. 2013
implementation of Hg in GAINS; phase-out of this technology is imminent, mandated through the Minamata convention, and no updates were necessary. Similarly, the methodology on estimating cremation emissions has been previously described in Rafaj et al. 2013. Activity projections for vinyl chloride monomer (VCM) production and Hg mining are only represented implicitly in the "OTHER_HG" sector. 2015 production values were assumed to be constant based on the data reported in the GMA'18, but phase-outs of the activity as mandated by the Minamata convention were applied to diminish activity projections.
The following mercury emission sources are not included in any GAINS sector for the current study: open savannah and forest burning, coal bed fires, and intentional mercury use in batteries, lamps, or other devices.

## 4 Scenarios

### 4.1 Control Strategies

Combined with the three energy and activity pathways listed in Section 3, scenarios of mercury control measures were devised.
They span the full range of possible anthropogenic mercury emissions to the atmosphere – from the complete absence of control measures – to current legislation, all the way to maximizing either co-benefit controls and applying stringent Hg-specific controls where this is feasible (see Table 2).

**Table 2: List of scenarios in this paper. The presented seven scenarios vary in two elements: Activity data incorporates assumptions**
**on energy and climate policy, pollutant control strategies incorporate different scenarios of clean air policy.**

| Activity data (Energy/Climate policy) | Pollutant control strategy (Clean Air policy) | Scenario ID |
|---|---|---|
| **Baseline scenario (BAS):** Adapted from the WEO 2022 Stated Policies (STEPS) scenario (IEA, 2022). Global energy demand growth met mostly by renewables; share of fossil fuels in global energy mix falls to < 75% by 2030, 60% by 2050. Global energy-related $CO_2$ emissions plateau at 37 Gt and fall to 32 Gt in 2050, leading to 2.5° C global average temperature rise by 2100. | **No control (NOC):** Hypothetical baseline of unabated emissions. No PM, $SO_2$ or Hg controls implemented. | 00_BAS_NOC |
| | **Current legislation (CLE):** Current legislation for Hg, PM and $SO_2$ | 01_BAS_CLE |
| | **CLE + Minamata scenario (MINA):** CLE for PM and $SO_2$; full implementation of Minamata BAT/BEP technologies and process phase-outs, as well as National Action Plans (NAPs) for ASGM | 02_BAS_MINA |





| | Co-benefit control for Hg, maximum feasible reduction for PM and $SO_2$ (co-MFR): Maximum co-benefits from PM and $SO_2$ for Hg emissions in all sectors; no additional Hg-specific controls. | 03_BAS_coMFR |
| | Maximum Feasible Reduction for Hg (Hg-MFR): Application of the most efficient Hg control implemented in the model for each GAINS sector. | 04_BAS_HgMFR |
| **Climate Policy scenario (CLIM1):** Adapted from WEO 2022 Announced Pledges (AP) scenario (IEA, 2022). Demand for all fossil fuels declines by 2030. $CO_2$ emissions fall to 12 Gt in 2050, leading to 1.7 ° C global average temperature rise by 2100. | **Current legislation (CLE):** Current legislation for Hg, PM and $SO_2$ | 05_CLIM1_CLE |
| **Net Zero scenario (CLIM2):** Adapted from WEO 2022 Net Zero Emissions scenario (IEA, 2022). $CO_2$ emissions fall to 23 Gt in 2030 before reaching 0 Gt in 2050, leading to < 1.5 ° C in 2100. | **Current legislation (CLE):** Current legislation for Hg, PM and $SO_2$ | 06_CLIM2_CLE |
| | **Maximum Feasible Reduction for Hg (Hg-MFR):** Application of the most efficient Hg control implemented in the model for each GAINS sector. | 07_CLIM2_HgMFR |

The **No Control scenario (NOC, as in 00_BAS_NOC)** represents the complete absence of control measures for any pollutant – unabated emission factors are displayed for all emission sources and all years. It displays higher than actual emissions, serving as a hypothetical end point that shows the efficacy of current legislation.

The **Current Legislation (CLE, as in 01_BAS_CLE, 05_CLIM1_CLE, 06_CLIM2_CLE)** control strategy represents existing and planned air pollution control policy for all pollutants implemented in GAINS (e.g. Rafaj et al. (2018), Amann et al. (2020) for global control strategies; Li et al. (2019) for China). Of these, the PM and $SO_2$ control strategies directly influence Hg emissions. In addition to co-benefit controls, Hg-specific controls were added in the extended modeling framework for this study; Hg-specific control measures were added for the cremation and waste incineration sectors in Europe and control

measures for non-ferrous metal production were adjusted to acid plants in line with existing legislation (e.g. the European Union Industrial Emissions Directive (Directive 2010/75/EU), Indian emissions regulations (CPCB 1998)). Table S7 in the appendix summarizes all Hg-specific control strategy changes.





For the **Minamata scenario (MINA, as in 02_BAS_CLE),** the CLE control strategy was extended by information from the available Minamata National Action Plans (NAP) for ASGM. Targets for good practice or elimination of mercury use in this sector were collected on the country level, then aggregated into the GAINS regional levels and then WEO regional level (Table S7)[5].

Maximum Feasible Reduction (MFR) scenarios.assume the implementation of currently available emission reduction technology that achieves the lowest airpollution emission factors. Such scenarios have been computed for pollutants including PM and $SO_2$ in GAINS using optimization procedures (e.g. Amann et al. (2011) and Wagner et al. (2013)). An MFR scenario with maximized PM and $SO_2$ controls, but CLE Hg controls generated. This scenario is called the **co-benefit MFR scenario (co-MFR, as in 03_BAS_coMFR)** and it simulates the maximum Hg reduction that can be achieved without Hg-specific measures, solely through co-benefits from air pollution policy.

Lastly, to demonstrate the end point of the control measures represented in GAINS, the Hg-specific **maximum feasible reduction scenario (Hg-MFR, as in 04_BAS_HgMFR, 07_CLIM2_HgMFR)** was generated. It represents the full application of the APCDs and Hg-specific control measures (or their combinations) with the highest removal efficiency for each sector and each GAINS region in 2050.  For combustion of coal, heavy fuel oil, diesel and waste, as well as most industrial processes including gold production these are Hg-specific control measures. In sectors with low emission factors where no Hg-specific controls are currently commercially applied such as road transport, biomass combustion domestic/residential fuel combustion, the co-benefit control with the greatest removal efficiency for Hg was applied. Activities where a ban is a viable policy option, such as ASGM, are banned. The only exceptions are Hg mining and VCM production, as they are implemented indirectly in GAINS (see the discussion in section 3 of 'HG_OTHER') and do not have control measures applied to them, but activities represent current Minamata policies. For the waste sector, the multi-pollutant waste management controls were applied, the most Hg-efficient of which is incineration, coupled with sorbent injection before an additional fabric filter (FFSINJ). Table S8 in the SI list the MFR control strategy in the year 2050.

## 4.2 Uncertainty of scenario results

Uncertainties for the aggregated sectors were estimated using a Monte Carlo Simulation approach. Uncertainties for uncontrolled emissions (scenario 00_BAS_NOC) were modelled by varying unabated emission factors and activity based on uncertainty estimates. The Monte Carlo Simulation was conducted by varying UEFs and activity data on the most granular GAINS resolution (182 regions, all sectors as listed in Tables S1, S10). The results were then aggregated to the regional and sectoral level used in this study.

---

[5] It is important to note that at the time of writing, not all ASGM-producing countries have published NAPs, meaning that likely, not all ASGM reduction targets can be represented in this scenario as of yet and that Hg reductions in this sector will likely be larger once all NAPs are published.





## 5 Results and discussion

### 5.1 The Baseline and current legislation scenario (01_BAS_CLE)

The global emissions trajectory for the baseline scenario of this study (01_BAS_CLE, as displayed in Table 3 and in Fig. 3,
295    compared with other control strategies), sees a slight increase in Hg emissions until 2050 to 109.6% of 2015 levels. Decreases
from power generation, residential combustion and a small decrease in non-ferrous metal production are offset by emissions
from waste and industrial emissions during both combustion and production processes. However, much of the increase is due
to the increase in reported gold production between 2015 and 2020, since it is assumed that gold production is  constant from
2020 to 2050, so 75% of the increase is due to ASGM. Similarly, emissions of Hg from waste sector increase as they are driven
by projected population increases (see Fig. 3). If these highly uncertain estimates are discounted, the combustion, metallurgy
and other processes sectors reduce their emissions slightly, by 80 t/year until 2050. On a regional level, these trends are largely
confirmed, but depend strongly on the dominating emission sectors and assumed controls in each region as plotted in Figure
4.

In **Europe**, emissions decrease by 18%, largely due to a reduction in Hg from combustion and industrial processes, followed
by transport and waste (OTHER). There is a significant relative shift in the dominant emission sector as well: the main emission
source become non-ferrous metals (63% of 2050 emissions) and cement (10%). For the NFME sector, there is not much
potential of emission reduction left, in the GAINS model as acid plants, the most efficient control technology currently
implemented, are already mandated[6]. A very similar trend can be observed for **North America**.

**African** Hg emissions are dominated by ASGM emissions, which remain constant at around three quarters of the total. Small
increases in cement and waste sectors point at a growing trend for population and resulting building activity, while combustion
emissions remain at the same level as currently. **Central and South American** emissions paint a very similar picture with
ASGM emissions being 84-85% of total emissions. For both regions, it is important to note that ASGM estimates for 2015 are
subject to large uncertainty (Keane et al. 2023) and projections in all scenarios can only show the influence of Hg policy such
as the Minamata convention, not reflecting the current forecast for production numbers of this sector.


---

[6] To further reduce Hg in NFME, the removal efficiency of acid plants would need to be increased. The GAINS removal efficiency of acid plants is significantly lower than their assumed removal efficiency in the GMA'18. This is due to the speciated emission accounting approach. The removal efficiency of PR_AP for $Hg^0$ is 91% (see Table A6 in the appendix and sources therein). As 80% of the emissions are assumed to be $Hg^0$, this leads to an overall removal efficiency of 92.7% as opposed to 99.98% in the GMA'18. Better data on emission speciation would be needed for a more exact estimate for this sector, and there might be an overestimation of European NFME emissions in GAINS.





**Table 3: Mercury emissions in the Baseline + Current Legislation (01_BAS_CLE) scenario by world regions and by sectors (tons year⁻¹).**

|  |  | ASGM | CEM | COMB_ IND | COMB_ OTHER | COMB_ POWER | GOLD | IND_ PROC | NFME | OTHER | Total |
|---|---|---|---|---|---|---|---|---|---|---|---|
| 2015 | **Global** | **952.61** | **153.69** | **85.20** | **99.27** | **263.21** | **43.60** | **110.09** | **349.69** | **263.24** | **2320.61** |
|  | Africa | 280.24 | 11.18 | 1.94 | 15.86 | 8.98 | 11.80 | 2.89 | 12.12 | 14.00 | 359.01 |
|  | Asia Pacific | 301.03 | 104.85 | 73.47 | 69.91 | 179.56 | 3.46 | 53.41 | 163.40 | 142.32 | 1091.41 |
|  | China | 48.27 | 61.76 | 46.87 | 47.97 | 68.19 | 0.69 | 33.07 | 85.66 | 93.02 | 485.51 |
|  | India | 6.37 | 21.91 | 18.82 | 14.88 | 92.04 | 0.00 | 9.82 | 8.12 | 25.46 | 197.43 |
|  | Japan | 0.00 | 1.89 | 0.69 | 0.22 | 2.04 | 0.03 | 3.19 | 7.62 | 4.63 | 20.33 |
|  | Southeast Asia | 240.91 | 12.08 | 2.99 | 3.16 | 6.38 | 0.44 | 3.01 | 14.00 | 11.17 | 294.13 |
|  | Central & South America | 352.80 | 5.46 | 2.25 | 1.84 | 2.36 | 7.32 | 7.31 | 18.60 | 12.99 | 410.93 |
|  | Brazil | 50.27 | 0.81 | 1.46 | 0.65 | 0.74 | 0.27 | 3.92 | 6.20 | 6.42 | 70.73 |
|  | Eurasia | 14.31 | 3.90 | 1.42 | 1.84 | 10.09 | 14.60 | 11.90 | 55.51 | 17.17 | 130.73 |
|  | Russia | 7.57 | 1.43 | 0.85 | 0.96 | 8.68 | 8.55 | 10.09 | 35.81 | 15.19 | 89.12 |
|  | Europe | 0.00 | 13.14 | 2.46 | 7.67 | 35.33 | 1.31 | 19.89 | 81.90 | 20.18 | 181.89 |
|  | EU 27 | 0.00 | 6.97 | 1.54 | 4.88 | 26.43 | 0.42 | 15.17 | 78.21 | 11.40 | 145.02 |
|  | Middle East | 0.23 | 12.39 | 0.69 | 0.30 | 1.43 | 0.29 | 6.28 | 4.07 | 27.50 | 53.18 |
|  | North America | 4.00 | 2.78 | 2.96 | 1.84 | 25.47 | 4.82 | 8.42 | 14.09 | 29.08 | 93.45 |
|  | USA | 0.00 | 1.78 | 2.41 | 1.13 | 24.25 | 0.85 | 5.52 | 1.44 | 20.08 | 57.45 |
| 2050 | **Global** | **1130.62** | **223.51** | **131.57** | **42.27** | **127.90** | **25.84** | **115.38** | **337.23** | **319.12** | **2598.27** |
|  | Africa | 333.84 | 25.55 | 4.14 | 16.79 | 2.30 | 1.53 | 4.81 | 1.83 | 29.74 | 440.24 |
|  | Asia Pacific | 369.21 | 150.76 | 115.59 | 14.85 | 108.09 | 3.52 | 73.04 | 202.59 | 139.68 | 1251.32 |
|  | China | 44.21 | 63.73 | 34.45 | 6.88 | 49.64 | 0.63 | 31.76 | 91.05 | 55.53 | 377.87 |
|  | India | 8.72 | 51.63 | 62.09 | 3.80 | 30.58 | 0.00 | 26.52 | 19.18 | 43.37 | 245.90 |
|  | Japan | 0.00 | 1.79 | 1.14 | 0.18 | 0.55 | 0.03 | 2.87 | 7.14 | 4.47 | 18.17 |
|  | Southeast Asia | 310.17 | 18.33 | 6.88 | 1.18 | 13.61 | 0.29 | 7.27 | 21.83 | 23.98 | 403.56 |
|  | Central & South America | 401.99 | 10.35 | 3.65 | 2.66 | 1.19 | 6.05 | 6.63 | 15.78 | 24.27 | 478.83 |
|  | Brazil | 55.99 | 1.09 | 2.50 | 1.84 | 0.64 | 0.30 | 3.46 | 5.37 | 13.30 | 84.49 |
|  | Eurasia | 22.00 | 4.50 | 2.45 | 0.82 | 7.56 | 7.72 | 7.16 | 2.14 | 18.81 | 94.55 |
|  | Russia | 9.14 | 1.65 | 2.04 | 0.28 | 4.80 | 1.18 | 6.15 | 2.14 | 16.79 | 44.16 |
|  | Europe | 0.00 | 14.53 | 2.98 | 2.69 | 6.91 | 1.84 | 11.03 | 95.11 | 13.17 | 148.26 |
|  | EU 27 | 0.00 | 8.06 | 1.82 | 2.32 | 0.77 | 0.52 | 5.31 | 89.19 | 6.04 | 114.04 |
|  | Middle East | 0.29 | 13.34 | 0.57 | 0.26 | 1.41 | 0.01 | 5.74 | 2.08 | 61.74 | 85.44 |
|  | North America | 3.29 | 4.48 | 2.21 | 4.21 | 0.44 | 5.17 | 6.97 | 17.70 | 31.70 | 76.17 |
|  | USA | 0.00 | 2.44 | 1.72 | 3.31 | 0.31 | 0.79 | 4.32 | 1.90 | 17.84 | 32.63 |






**Figure 3: Global mercury emissions for the BAS scenarios and all different control strategies – by sector.**






**Figure 4: Regional data: Hg emissions for the Baseline, Minamata, co-MFR and BAS_HgMFR scenarios (01_BAS_CLE, 02_BAS_MINA, 03_BAS_coMFR, 04_BAS_HgMFR. (a) Africa (b) Central and South America, (c) Middle East, (d) Europe, (e) Eurasia, (f) North America, (g) Asia Pacific, (h) China, (i) India, (j) Southeast Asia.**

The **Asia-Pacific** region spans the majority of the world population and produces the majority of the world's Hg emissions. Emissions represent 46% of global emissions in 2015 and 48% in 2050. Globally, most emissions from power generation and combustion come from this region. Emission reductions are projected in the Baseline (01_BAS_ CLE) only in combustion in power plants, as well as residential combustion. Emissions related to manufacture and building, such as cement production, industrial processes, NFME smelting and also population-related releases such as waste increase. ASGM emissions are regionally highly variable. Within the Asia-Pacific region, large differences pervade on the country and subregion level (Fig.



4). Emissions in China are projected to decrease in the Baselinedue to both decarbonization and co-benefits from rapid and
stringent application of PM and SO₂ controls. Indian Hg emissions from fossil fuel combustion (in COMB_POWER) are
projected to increase. ASGM emissions dominate in Southeast Asia. Japan is a typical example of an industrialized country –
there is a decrease in combustion emissions to 2050, but a stable trend in NFME emissions, as there is little scope for reduction
left within the air pollution control policies, except for lowering production volumes.

Emissions from the **Middle East** are dominated by cement production and unmanaged landfill waste emissions.

**5.1.1 Uncertainty**

Table S12 presents the percent ranges for the aggregated emission sectors on a global level. ASGM emission variations are
largest on a relative level, but due to the large uncertainty in emission factors and the high unabated emission factors in the
NFME sector, the absolute range of emissions was largest in this sector, followed by OTHER emissions and CEM. Combustion
sectors show a small spread, reflecting the good data quality. While large-scale gold emissions give a large relative spread in
the upper and lower bounds, their low total emissions mean that their contribution to the overall uncertainty of the results is
small. Assuming that the abated emission factors, which take into account Hg emission reductions through abatement
measures, have the same levels of uncertainty as the UEFs, the calculated relative uncertainties are applied to 01_BAS_CLE
(see Figure S1). The result is that the main uncertainties of this model lie within the ASGM sector; ASGM emissions make up
between a third and 47% of total emissions, looking at the lower and upper range estimates, respectively, varying between 602
and 1373 t in 2015. After this, Waste emissions have the most variability in absolute terms, followed by emissions from the
non-ferrous metal and power sector. To conclude, data quality in ASGM and non-ferrous metals and waste, as well as cement
will significantly improve the overall quality of the baseline mercury emissions estimates.

**5.2 Mercury emission reductions from PM and SO₂ co-benefits**

Comparing the No Control and Baseline (BAS_CLE) scenarios, the full extent of emissions reductions from current clean air
policy becomes apparent, as NOC emissions are more than double those of the Baseline in 2015 illustrating that already, more
than 50% of potential unabated Hg emissions are avoided through clean air policy. An extended discussion can be found in
section S4 and Fig. S1 of the appendix.

The further potential for Hg emission reductions through co-benefits from PM and SO₂ policy are assessed by comparing the
Baseline with a co-MFR case. It shows the total Hg emission reduction potential from PM and SO₂ abatement, without
considering any Hg-specific measures beyond those already implemented in the CLE case. In 2050, the co-MFR scenario
projects 2023 t emissions compared to 2455 t in the Baseline – a reduction of 18%.

In most of the world's regions, the comparison of co-MFR compared to CLE scenarios reveal that large capacities are already
controlled by at least PM controls and some form of SO₂ control policy is already put in place for 2050, leading to



implementation of the Hg control measure 'PM_FGD' in the power sector in GAINS - the most efficient co-benefit control
       technology. An extended discussion of the technology shares can be found in the SI, section S3.2. In China alone, the
       retrofitting measures for coal-fired power plants in the 12th Five-Year-Plan from 2010 to 2015 have reduced Hg emissions by
       23.5 tons, explaining why further reductions in the power sector are limited as current policy already mandates the strictest
       levels of APCD deployment, closure of inefficient small plants (e.g. Li et al. 2020) and coal phase-out policies. On the other

hand, there is still scope for improvement in the industry sectors such as NFME, cement, gold and other production. In 2050,
       emissions in the co-MFR for China are only 6.2% (22.5t) lower than in Baseline. The difference is mainly found in the OTHER
       (17.4 t) and IND_PROC (4.6 t) sectors, indicating that co-benefit PM and $SO_2$ control measures are already planned to be
       maximized in the currently active policy. The results for India imply that co-benefits from air pollution policy can still make
       significant contributions to lower Hg emissions, totaling 12.4% reduction in the co-MFR case compared to the Baseline. The

largest reductions are 18.0 t Hg reduction in cement production in 2050 between the Baseline and the co-MFR case and 10.1 t
       Hg reduction in OTHER, with contributions mostly from waste management (see Fig. S3). In the Middle East, the potential
       for emission reductions through better waste management becomes especially clear (Figs. 3, 4, S3), as waste emissions in
       OTHER dominate the picture. In the Baseline, unmanaged waste allows Hg to be emitted into the atmosphere and waste
       generation is expected to increase with population growth. However, in the co-MFR scenario, these emissions are minimized

by 2050 thanks to an application of waste incineration with efficient Hg capture, the OTHER sector alone causing a 69.2%
       reduction in total Hg emissions in 2050 in the co-MFR compared to the Baseline. In North America, Europe including the EU
       27 and the Asia Pacific region, the most significant co-benefit reduction potential lies in the NFME sector, owing to its
       exceptionally high emission factors, where even small improvements in the co-benefit controls are able to cut tons of Hg
       emissions. For example, in the EU27, over 90% of the emission reduction is reached only in the NFME sector, which could

see a further 54.1 t Hg reductions EU-wide if the strictest Acid Plant controls are employed at all plants.

**5.3 The Maximum Feasible Reduction scenario**

Comparison with a maximum feasible reduction (MFR) scenarios can serve to quantify the maximum potential of targeted Hg
abatement (04_BAS_HgMFR, 07_BAS_HgMFR, short Hg-MFR) relative to other approaches. Here, the results of

BAS_HgMFR are discussed relative to the co-MFR scenario, which uses only air pollution co-benefit measures for Hg
       reduction (co-MFR, discussed in section 5.2.2). Both scenarios represent end points of possible policy developments, so this
       comparison is only made for the year 2050. Co-benefits from PM and $SO_2$ control impact mainly the power, industry and waste
       sectors, however, there are more efficient technologies available targeting Hg emissions applicable to a large range of sectors.
       For industry and combustion sectors where Hg-specific measures are expected, the Hg-MFR scenario assumes adoption of the

most efficient pollution control measures available, which in most sectors is sorbent injection in front of a fabric filter (FFSINJ)
       The Hg-MFR scenario implements a complete, global ban on ASGM and is thus the only scenario where the 1130 t of ASGM
       Hg emissions are projected to disappear, reducing total Hg emissions in 2050 drastically by 79% compared to the Baseline,





from 2455 t to 521 t. As seen in Figs. 3 and S3, the difference in 2050 emissions is also large between the Hg-MFR and the co-MFR: the co-MFR projects 2050 emissions to be 2023 t, 3.9 times higher compared to the 521 t in Hg-MFR. Even when

discounting the fact that ASGM emissions are completely removed from the Hg-MFR, and when comparing all sectors except ASGM, the co-MFR would still produce 371 t annual Hg emissions more than the Hg-MFR case. Apart from ASGM – the largest absolute reduction of 135 t can be achieved in the cement sector, where emissions in Hg-MFR are just 24% of those in the co-MFR scenario. This is followed by emissions from industrial processes, which, in 2050, are halved to 43 t in the Hg-MFR compared to 86 t in co-MFR. Little to no further emission reductions are estimated in the COMB_OTHER, NFME and

OTHER sectors. Waste emissions (from OTHER) are captured by multi-pollutant controls. This is especially visible in the Middle East region, where the majority of emissions is modeled to come from waste (in OTHER), as shown in the CLE and MINA scenarios (e.g. Fig. 4). This problem is solved in the co-benefit MFR scenario, where better landfill practices are adopted and the loosely managed waste is re-directed into waste incineration with mercury capture. Similarly, in NFME, the most advanced $SO_2$ controls, acid plants, are expected to already comply with strict Hg legislations and are expected to contain Hg-

specific sorbents, as sulfuric acid is sold on as a product and therefore needs to comply with the Hg limit values of this product. Combined with the Net Zero $CO_2$ policy, scenario 07_CLIM2_HgMFR presents the lowest-possible primary Hg emissions in 2050: Co-benefits from climate policy, clean air policy and Hg control policy are maximized and taken into account at the same time, leading to 446.7 t Hg emissions in 2050, a further 56.9 t are reduced in the combustion sectors relative to BAS_HgMFR (Figs. 3, 5).

**5.4 Mercury emission changes with climate policy**

To compare climate policy outcomes on Hg, the three energy scenarios (BAS, CLIM1 and CLIM2) with CLE controls (BAS_CLE, CLIM1_CLE, CLIM2_CLE) (see Fig. 5). Considering the large decline in $CO_2$ emissions by 2050 driven by climate mitigation goals simulated in the Stated Policies, Announced Pledges and Net-Zero Emissions scenarios (IEA 2022), our analysis suggests that the reductions in Hg emissions, while apparent in Fig. 5, occur at a significantly lower rate than for

$CO_2$.





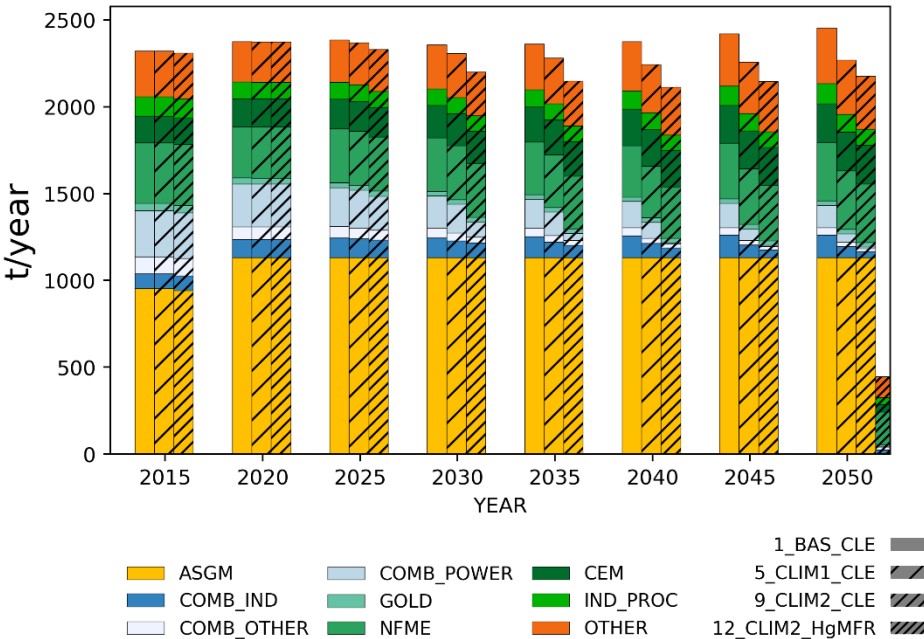

**Figure 5: Global mercury emissions for the three energy scenarios BAS, CLIM1 and CLIM 2 under the CLE control strategy – by sector.**

Noticeable emission reductions are projected in the combustion sectors, caused by a reduction in global consumption of coal,
oil, natural gas and traditional biomass use in the three scenarios. In contrast, emissions from cement production (CEM) and
'OTHER' (transport and waste) increase in all three scenarios between 2015 and 2050. While this doesn't offset the emission
reductions in CLIM1 and CLIM2, it leads to a slight increase in global emissions in the 01_BAS_CLE scenario. A shift of
emissions from the combustion sector towards industrial processes, gold production and waste treatment can be seen due to
changes in the energy and industrial systems and is most pronounced in scenario compliant with the most stringent climate
policy (Fig. 5, Panel b).

The decarbonization of global economy and transition towards renewable energy sources and associated infrastructure induces
an increased demand for critical minerals for electrification in all three scenarios. In GAINS, this is reflected as increased
demand for non-ferrous metals in 2050 (see Fig. 3, NFME activity). However, improved pollution control measures lead to
3.6% lower emissions in 2050 than 2015. A projected increase in activities and resulting emissions in all three scenarios relative
to 2015 are also shown in other industrial processes sector (IND_PROC), largely due to iron and steel production. In the
CLIM1 scenario, IND_PROC emissions dip to 93t in 2030, then increase again to 101t until 2050. Overall, the baseline (BAS)
scenario projects slightly rising emissions from IND_PROC, while both CLIM1 and CLIM2 show slight declines from 110 t/a
in 2015 to 101 t/a Hg (CLIM1) and 91 t/a (CLIM2) until 2050.

The largest differentiation between the different energy scenarios and their resulting Hg emissions is apparent within the
combustion sectors. Globally, emissions from the power sector (COMB_POWER) roughly halve from 2015 to 2050 in the





Baseline and drop further by 82% and 97% in CLIM1 and CLIM2 respectively, virtually removing the combustion sources of Hg from the emissions profile. It is noted that the largest reductions in Hg emissions occur in regions where coal power is a significant contributor to the energy mix. Globally, the largest relative share of combustion-related Hg emissions in 2050 is estimated within the industrial combustion sector. In the Baseline, emission levels from industries are projected to rise by 50%

between 2015 to 2050, however, in comparison these are reduced by a quarter in CLIM1 and by 56% in CLIM2.

While fossil fuel combustion in the power sector is projected to decline on a global level, in some countries such as India, fossil fuel combustion in industries (COMB_IND) is projected to grow beyond 2015 values in all scenarios, with Hg emissions projected to more than triple from 18.8 t/a in the Baseline to 62.1 t/a by 2050. In CLIM1, COMB_IND emissions rise by 7.7 t/a, and only in the net-zero CLIM2 scenario, Indian industrial combustion emissions decrease to 6.7 t/a by 2050.

Emissions of Hg from activities which are scaled by population growth, such as waste generation, rise equally in all scenarios. In the category 'OTHER', Hg emissions from transport are combined with emissions from waste and some other Hg emission activities. Consequently, waste emission increases are offset by emissions savings from fossil fuel-based road transport. Nevertheless, an overall increase in these emissions is seen across all three scenarios, although slightly higher in the Baseline than in the climate scenarios.

**5.5 Implications of Minamata convention on future Hg emission trends**

For the purpose of direct comparison between the modelling results and those Minamata Convention on Mercury (MCM) concerning Hg emissions to the atnosphere, a sector mapping between the MCM provisions and GAINS was conducted, as detailed in Table 4: GAINS sectors which are also included in the MCM are grouped into Annex B, C and D sources. Smaller emission sources in GAINS including some industrial processes, combustion of biomass, liquid and gaseous fossil fuels, as

well as transport emissions are not covered by the MCM[7]. Figure 6 shows the results of the BAS scenarios (CLE, MINA, coMFR, HgMFR), mapped to the Minamata sectors.

**Table 4. Source and process categories covered by Minamata Convention provisions, with their corresponding GAINS sector representation in the 'Minamata' sector aggregation.**

| Source and process categories | | Convention provisions | GAINS representation |
|---|---|---|---|
| Extraction and use of fuels/energy sources | | | |
| | Coal combustion in power plants | Article 8, Annex D | COMB_POWER |
| | Coal combustion in coal fired industrial boilers | Article 8, Annex D | COMB_IND |
| Primary (virgin) metal production | | | |

---

[7] It is noted that there are also emission sources which the MCM covers, that are not explicitly represented in GAINS. This concerns emissions from acetaldehyde production, other chemicals and polymers production, incineration of medical waste, hazardous waste and sewage sludge.



| | | | |
|---|---|---|---|
| | Gold (and silver) extraction with mercury amalgamation processes | Article 7, Annex C | ASGM |
| | Gold extraction and initial processing by other methods | Article 8, Annex D | GOLD |
| | Zinc extraction and initial processing | Article 8, Annex D | NFME |
| | Copper extraction and initial processing | | |
| | Lead extraction and initial processing | | |
| Cement production | | | |
| | Cement and clinker production | Article 8, Annex D | CEM |
| Intentional use of mercury in industrial processes | | | |
| | Chlor-alkali production with mercury-technology | Article 5, Annex B | OTHER |
| | Vinyl chloride monomer production with mercury catalyst | Article 5, Annex B | OTHER |
| | Acetaldehyde production with mercury catalyst | Article 5, Annex B | not in GAINS |
| | Other production of chemicals and polymers with mercury | Article 5, Annex B | not in GAINS |
| Waste incineration | | | |
| | Incineration of municipal/general waste | Article 8, Annex D | OTHER |
| | Incineration of hazardous waste | Article 8, Annex D | not in GAINS |
| | Incineration of medical waste | Article 8, Annex D | not in GAINS |
| | Sewage sludge incineration | Article 8, Annex D | not in GAINS |
| Other emission sources | | | |
| | E.g. transport emissions, various industrial processes, domestic coal burning | Not covered | NOT_COVERED |




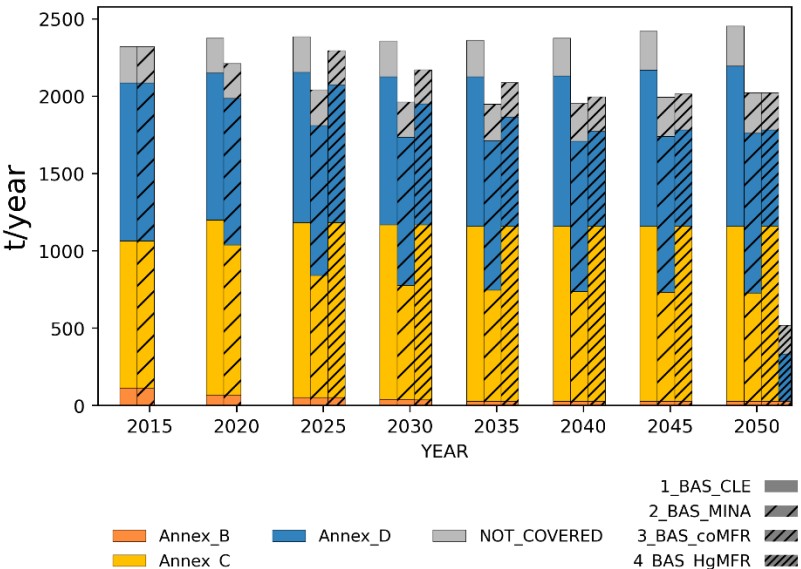

**Figure 6: Emissions from sectors covered by the Minamata convention.**

### 5.5.1. Annex D sources

Article 8 of the MCM addresses atmospheric emissions directly by '*controlling and, where feasible, reducing emissions of*
*mercury and mercury compounds, often expressed as "total mercury", to the atmosphere through measures to control*
*emissions from the point sources falling within the source categories listed in Annex D*' (*Minamata Convention on Mercury*,
2013, art. 8.1). Annex D sources are represented well in GAINS (Table 4). Article 8 does not mandate a phase-out of these
activities, but rather mandates use of BAT/BEP for new sources and it is up to the parties of the MCM to formulate appropriate
steps to reduce Hg emissions from existing sources – which may be co-benefit reduction of Hg from other air pollution control
or Hg-specific measures. The different control strategies in the CLE, co-MFR and Hg-MFR scenarios (01_BAS_CLE,
03_BAS_coMFR, 04_BAS_HgMFR) represent different narratives of how the art. 8 objectives could be achieved, as different
Hg reduction measures may be considered BAT/BEP. For example, the EU's Industrial Emissions Directive (EU-IED) states
for large coal and lignite combustion plants that PM, $SO_2$ or NOx as well as Hg-specific reduction technologies are considered
BAT (see BAT 27 in Commission Implementing Decision (EU) 2021/2326 (2021)). Fuel choice is also listed as a Hg-specific
control measure but might be considered a co-benefit from energy and climate policy in the case of Hg emissions from coal.
Thus, each scenario (Baseline, co-MFR, Hg-MFR, CLIM1_CLE and CLIM2_CLE, and combinations thereof) would be
compliant with Art. 8 for coal combustion. For non-ferrous metals, BAT is to consider raw materials with low Hg contents, as
well as using Hg-specific sorbents in the EU (BAT 11 in Commission Implementing Decision (EU) 2016/1032 (2016)), the
end point of which are represented in the Hg-MFR scenario. For cement, the EU BAT conclusions are a combination of limiting
Hg content in the feed materials, as well as utilizing dust control co-benefits (BAT 43 and 54, Commission Implementing
Decision (EU) 2013/163/EU (2013)), represented in the CLE and co-MFR scenarios. Dedicated Hg controls are required in



waste incineration (BAT 31, Commission Implementing Decision (EU) 2019/2010 (2019)) which would represent a Hg-MFR scenario if implemented globally. Annex D sources are globally dominated by NFME and coal power emissions in 2015, but power sector emissions are expected to fall in importance until 2050, as discussed earlier. Industrial emissions (NFME, CEM) as well as waste emissions are projected to become the largest contributors to total emissions in Annex D in all scenarios. The present results suggest that targeted, Hg-specific BAT recommendations result in the most efficient Hg reduction for Annex D sources.

### 5.5.2. Annex C sources

Article 7 does not exclusively address air emissions but concerns the phase-out of ASGM activities: Each party with ASGM activities on their territory '*shall take steps to reduce, and where feasible eliminate, the use of mercury and mercury compounds in, and the emissions and releases to the environment of mercury from, such mining and processing.*' (*Minamata Convention on Mercury*, 2013, art. 7.3). The National Action Plans (NAPs) required by Art. 7 lay out the planned reduction / phase-out for each relevant party. Where NAPs are already published, the reduction targets have been implemented in the BAS_MINA scenario. With already published targets, ASGM emissions in BAS_ MINA are projected to sink to 738.3 t in 2030 and 700.0 t in 2050 and ASGM is bound to make up roughly a third of global Hg emissions in 2050, the largest absolute reduction stemming from Southeast Asia (272.0 t) and Africa (116.3 t). However, there are large uncertainties connected to activity levels as well as emission factors of ASGM (determined to be -37% to +44% error in this study's Baseline on a global level).

### 5.5.3 Sources not covered by the Minamata Convention

Further, it is important to note that the MCM does not cover all potential emission sources and not all mercury uses, instead focusing on intentional use of Hg and Hg compounds, as well as the largest emitters. In this study, only 10% of the emissions computed by GAINS fall into the 'NOT_COVERED' category, highlighting that the MCM has the potential to impact 90% of the emissions shown in this paper.

### 5.6 Speciation

The speciation of Hg emissions to the atmosphere strongly influences their fate and spatial distribution (e.g. Selin 2009). In this study, $Hg^0$ dominate in all scenarios, making up between 76% and 90% of total emissions in the year 2050, as can be seen in Fig. 7. The scenarios with Hg-specific control measures (BAS_HgMFR, CLIM2_HgMFR) project the lowest proportions of $Hg^0$ emissions with 76% and 77%, respectively, while the scenario BAS_coMFR, which leans most heavily on co-benefit control of Hg displays the highest share of 90% $Hg^0$ emissions. This is due to the fact that the PM and $SO_2$ controls implemented in GAINS tend to have higher removal efficiencies for $Hg^{II}$ and $Hg_p$, while the harder-to-abate $Hg^0$ requires targeted approaches as implemented in the Hg-MFR scenarios. As ASGM emissions are assumed to be only elemental $Hg^0$, scenarios with higher ASGM abatement such as BAS_MINA has a lower share of $Hg^0$ emissions than BAS_CLE (84% and 87%, respectively).



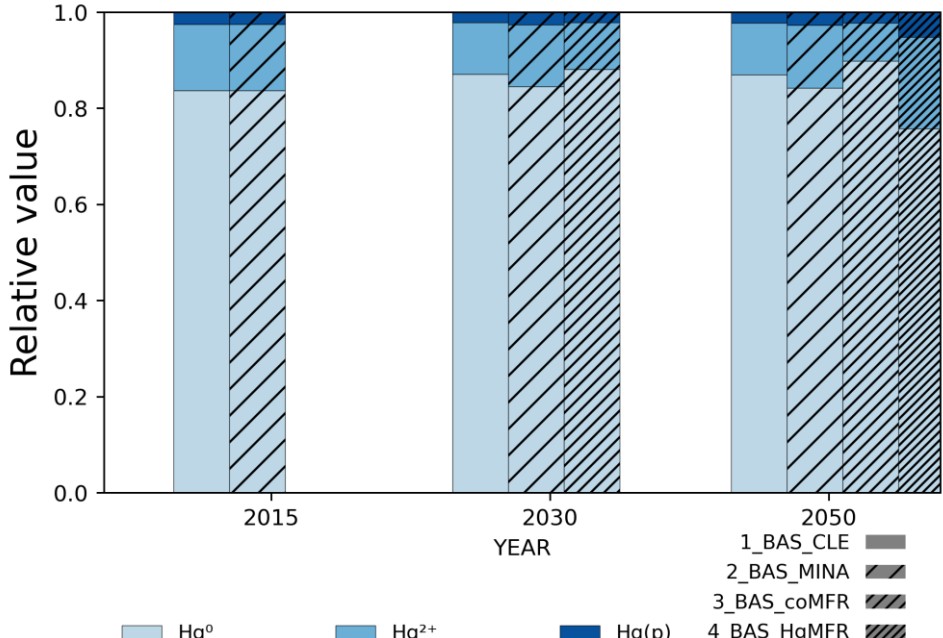

**Figure 7: Share of mercury species in different scenarios.**

**5.7 Further work**

While the future scenarios are indicative of the abatement potential of different types of policy, there are some simplifications built into the current implementation of Hg within the GAINS model that would warrant further attention and present areas of future work.

(1) When considering future projections of **non-ferrous metal smelting**, the share between Cu, Pb and Zn and the proportions of primary vs. secondary mining within the NFME category is fixed to the 2015 level, not dynamic. This is a simplification that can lead to an overestimation of Hg emissions from this sector, as copper production generates relatively lower Hg emissions compared with lead and zinc, but its share in the NFME category is projected to increase as it is a critical metal in the decarbonized economy.

(2) Similarly, assumptions on Hg levels in the **waste sectors** have one fixed, global unabated emission factor, derived from total estimates of emissions from waste in 2015. These numbers are projected into the future using shares of different waste types, as well as population and macroeconomic projections. However, again, the fixed emission factor might change, as Hg policy reduces Hg levels in consumer products. Furthermore, the emissions are likely to be heterogeneous on a regional scale. Better projections of Hg in wastes will increase the accuracy of waste emissions estimates as well can better simulate shifts in waste composition under assumptions of circular economy.



(3) The Hg removal efficiency of **NOx controls** could be studied further and NOx control policy and their interplay with Hg, PM and SO$_2$ control technologies could be included into the GAINS algorithm, thus making the calculation of removal efficiencies more detailed. This is especially relevant for the consideration of Hg speciation, as selective catalytic reduction technology for NOx removal from flue gas systems can be optimized to oxidize Hg$^0$ to Hg$^{II}$, thus changing the share of species

as well as increasing the efficiency of particle filters and flue gas desulfurization units in keeping mercury from the atmosphere (Usberti et al. 2016).

(4) Projections of **artisanal and small-scale gold mining** into the future need to be improved as additional data and a better understanding of the ASGM drivers are emerging from the Minamata Convention process and the scientific community. Similarly, policy measures for ASGM emission reduction might be refined and added in GAINS.

(5) As is done for traditional pollutants, optimization would allow the calculation of cost-optimal meeting of reduction targets. Further, GAINS model output could be used as input to dispersion modelling, and the subsequent calculation of health and environmental impacts. The authors are also working on making some of the Hg scenarios accessible to the public via the GAINS online tool (include the link).

**6 Conclusions**

This study explores future anthropogenic mercury emissions through seven scenarios combining different energy and climate strategies (BAS, CLIM1, CLIM2) with policies to abate mercury as well as traditional pollutants such as PM and SO$_2$ (CLE, MINA, co-MFR, Hg-MFR). The Baseline (BAS_CLE) projects a slight increase of 5.7% in global emissions in 2050 compared to 2015, despite a 32.6% reduction of emissions from combustion, due to increased cement and other emissions including waste (+45.4% and +21.4% resp.). The comparison of three climate scenarios under current legislation for clean air policy

shows that the Hg emission reduction from the fossil fuel combustion sectors depends on the level of climate policy ambition, which prompts a transformation in the energy system towards non-coal sources, enabling a range of 30% (BAS_CLE) - 86% (CLIM2_CLE) emission reduction. However, there are trade-offs such as a 45% increase in cement emissions.

In all studied sectors, emission increases can be dampened or reversed by conventional air pollution control measures. High levels of co-benefits of PM and SO$_2$ control are already present in the current legislation, as comparison to the No Control

Scenario demonstrates. Compared to the Baseline, a 17.6% reduction in emissions can be achieved in 2050 solely through maximizing PM and SO$_2$ control (BAS_coMFR). This strategy is especially effective strategy for the waste sector (represented in OTHER), where emissions fall by 58.9% compared to the Baseline in the 2050. Drastic reduction of ASGM as set out in the Minamata Convention is only possible with a Hg-focused approach, as the MINA and Hg-MFR scenarios show. ASGM offers the largest absolute Hg reduction potential of any sector, but also the largest uncertainty in the baseline emission

estimate. OTHER emissions (including waste) are a large source of uncertainty in this study, and projections range between a 22% increase and 54% decrease in 2050 depending on both climate and clean air policy.

The Minamata convention covers roughly 90% of the Hg emissions computed by the GAINS model. Annex D sources of the convention are regulated through BAT/BEP, which oftentimes means co-benefit pollution control, as set out in the co-MFR scenario, so the CLE, co-MFR and Hg-MFR scenarios represent three different narratives that could represent pathways of

different countries to reduce Hg from Annex D sources.

Overall, the findings emphasize the importance of implementing targeted Hg control policies in addition to stringent climate, PM and SO₂ policies to achieve significant reductions in Hg emissions.

*Code and data availability.*

The aggregated data discussed in this paper is in the electronic supplement. The GAINS model can be accessed upon

registration through the online interface (http://gains.iiasa.ac.at/gains/GOD/index.login).

*Author contributions*

Conception of and study design … PR, FB. Concept and methodology … FB, PR, FB, FW. Implementation into the GAINS model … FB, RS. Data collection, uncertainty calculation, preparation of manuscript … FB. Manuscript review and supervision … PR, JMJ, FW.

*Competing interests.*

The authors declare no competing interests.

*Acknowledgements*

This work has been supported by the IIASA Young Scientists Summer Programme 2021, as well as by the RC-UK Centre for Doctoral Training in Bioenergy [Grant number EP/L014912/1]. FB thanks Zbigniew Klimont for welcoming and supporting

the project, as well as comments on the manuscript.

*Supplementary Information.*

The following files were submitted separately:

2023-ACP_SI_ScenarioData.xlsx

2023-ACP_SI_Tables.docx



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
