# Peer review of "Global scenarios of anthropogenic mercury emissions"

_EGUsphere, 2024_

## Author Comment (AC1)

**Response to Reviewers**

**April 5, 2024**

**Manuscript EGUSPHERE-2024-41**

**Title :** "Global scenarios of anthropogenic mercury emissions"

**Corresponding Author:** Flora Maria Brocza, brocza@iiasa.ac.at

**Response to Reviewer #1**

*This is a very needed, detailed, and well-documented paper developing new Hg scenarios (and an underlying model) that is sorely needed in the community. The underlying work is of high-quality and ACP is an appropriate journal for it.*
*Before publication, however, there are two substantial issues that should be addressed.*

**General Comments**

**Comment 1:** *There are several substantive errors and misstatements on the requirements of the Minamata Convention, some of which are just in the writing, but some of which also percolate into the analysis and the technical analysis. These are listed below in detailed comments. The most significant conceptual one is that scenarios are defined as BAT/BEP scenarios, but as far as I can tell this doesn't actually follow the Minamata definition (which depends on economic considerations and qualitative feasibility concerns which aren't taken into account in the modeling done here). This should be clarified, especially where BAT/BEP is used to imply mercury-specific controls (where in many regions, countries may define BAT/BEP as co-benefits in practice, especially in the near term). This makes the discussion of implications a bit more difficult, but is fixable I think with some careful attention to language and examination of the degree to which this might affect the underlying analysis.*

**Reply 1:** Thank you for the close reading of this manuscript with regards to the Minamata Convention and its specific provisions!

We have re-read all text passages containing references to BAT/BEP technologies, either with reference to Minamata or European legislation. We have also referred back to the original Convention text, especially Art. 5(d), and additional BAT/BEP guidance. In view of this, we have split our response into two parts:

(1) "scenarios are defined as BAT/BEP scenarios"
We agree that either co-benefits or Hg-specific technology may become BAT/BEP in different parties to the Minamata Convention, and, at this point, it is not known to us from official documents which measures pursuant to Art. 5(d) a party will choose to implement to reduce emissions from their Annex D point sources. Thus, as you point out, it would be difficult to define a global "BAT/BEP" scenario. It was not our intention to provide such a scenario, but we see that the **description of 02_BAS_MINA in Table 2 might be the source of the misunderstanding**. It was meant to be just another way of stating that 'co-benefits for PM and SO2 arising from current legislation' are implemented into the scenario, but in light of your comments, this is simply incorrect and has been deleted. In Lines 277 ff. and Table S7, scenario construction is detailed further, explaining that only quantifiable Minamata goals from the NAPs were included.

*(2) [...] where BAT/BEP is used to imply mercury-specific controls (where in many regions, countries may define BAT/BEP as co-benefits in practice, especially in the near term).*

We have not undertaken a technology-specific interpretation or generalization meaning of BAT/BEP in the context of the MCM. One notable exception to this might be waste incineration and cremation in Europe, where we have assumed full implementation of Hg-specific sorbent injection before a fabric filter (FFSINJ in GAINS jargon), in line with the low emission limit values and BAT laid out in European legislation.

*(3) This should be clarified*

The following line has been added:

Line 280 f.:

> It is important to note that this scenario does not include any assumptions on the implementation of emissions reductions pursuant to MCM Article 8 beyond co-benefits from current legislation.

Line 497 in Section 5.5.1 has been clarity-edited:

> Thus, each scenario (Baseline, co-MFR, Hg-MFR, CLIM1_CLE and CLIM2_CLE, and combinations thereof) would represent a variation of possible compliance with Art. 8 for coal combustion.

Line 503 f. in Section 5.5.1

> The scenario results suggest that targeted, Hg-specific measures result in the most efficient significant Hg reduction for Annex D sources, as they are generally associated with lowest emission factors. .However, after taking into account region-specific and economical factors, this may not translate into BAT/BEP for many countries and represents and represents an end point of lowest possible emissions from these sources.

L. 589 in Section 6 (Conclusions)

> The Minamata convention covers roughly 90% of the Hg emissions computed by the GAINS model. Annex D sources of the convention may be regulated differently by each convention party and the CLE, co-MFR and Hg-MFR scenarios combined with different climate policy delineate the option space of possible Hg reductions from these sources.

**Comment 2:** *The climate scenarios included are (somewhat infeasibly) optimistic. This is fine in principle, but uncommented on (at least the paper should note that these are very low C scenarios); plus, it might be more useful for policy-making for a more realistic set of scenarios to be available.*

**Reply 2:** Thank you, indeed the mitigation scenarios included are probably overly ambitious. Both are GAINS implementations of IEA World Energy Outlook (WEO) scenarios, i.e. we have not developed them ourselves, but rather just represent them in GAINS. We have chosen to use such global scenarios because we believe that the WEO scenarios offer orientation for a broad range of audiences. The Baseline scenario considers a detailed set of current and planned country- and sector-specific policies analyzed by IEA as of the year 2022, while the CLIM1 case takes further into account the updated climate commitments made by governments including NDCs as well as longer term net zero emissions targets pledged at the Glasgow COP26 (see https://www.iea.org/policies for further reference). When compared to other similar scenarios reported by IPCC AR6, it is relatively optimistic in terms of efficiency improvements, deployment of renewables and emerging fuels (H2), on the other hand it is much more conservative on the projected role of CCS and carbon removals. In addition, since the publication of WEO2022, some economies have announced even more ambitious climate targets, e.g., the EC proposes a commitment to reduce the net EU GHG emissions by 90% by 2040 compared to 1990 (https://climate.ec.europa.eu/eu-action/climate-strategies-targets/2040-climate-target_en).

We agree that overly ambitious scenarios are not directly useful for policy making if the expectation is that policy makers are prepared to directly translate scenarios published in an academic journal into policy. However, scenarios help to delineate option spaces. Our experiences with policy makers suggest that very ambitious but technically feasible scenarios are useful to describe the extreme ends of the feasibility space and thus provide context. Very ambitious mitigation scenarios also help to estimate the maximum potential for co-benefits on emissions of air pollutants, as well as

estimates of residual emissions. That said, one of our main takeaways from this study is that in all scenarios, Hg emissions from combustion become relatively less important compared to other emission sources, and dwindle when compared to ASGM emissions – and that co-benefits from climate policy will ultimately not solve the Hg pollution problem.

In summary, we have now added text indicating that the climate scenarios used are indeed ambitious:
L. 228 f.:
> Although both climate scenarios are very if not unrealistically ambitious, they provide a scope to quantify a Hg-reduction potential induced through a rapid decarbonization of the global energy system.

**Comment 3:** *I don't think either of these changes require a lot of additional work (although I'd argue that showing a climate policy scenario that's somewhat less infeasible would really add value). But they are substantive, and the paper as it stands really needs to be corrected. Finally, I would recommend that some edits to be made to the discussion in section 5 to make the take-home messages clearer and more apparent to the reader.*

**Reply 3:** Thank you, the respective sections have been revised/reedited:
- Any passages containing reference to BAT/BEP provisions have been clarity-edited (see Reply 1)
- Several clarity edits have been made to Section 5, especially Section 5.5.1. (Annex D sources), Section 5.5.3 (Sources not covered by the Minamata Convention).
- Section 6 (Conclusion) has been edited to serve better as a concentrated take-home message.

**Specific comments:**
*Line 8*: this is incorrect, the Minamata Convention does not ban mercury trade. Please refer to Article 3 -- there is a Prior Informed Consent procedure. Relatedly, it would be more accurate to say that the Convention bans certain mercury uses, or specific mercury uses (here, the language could be interpreted to apply to all Hg uses)

Done. Thank you for pointing this out. Some detailed context has been lost here in the interest of brevity, which is of course not acceptable.

*Lines 8-10*: these "co-benefits" are specifically set out for in the Minamata Convention, please see Article 8 (5d) and related BAT/BEP guidance

Thank you for this comment. Please also refer to Reply 1 for a wider discussion of this issue and changed text passages. 'Co-benefits' are discussed in the wider scientific context for any measures which might cause a reduction on Hg emissions although their intended target is another pollutant, such as greenhouse gases, PM, SO2 or NOx. If co-benefits from current legislation would not be considered for all sectors and regardless of MCM provisions, our computed emissions would not be correct.
That said, the definition of co-benefits for Annex D sectors as set out in the Minamata BAT/BEP guidance are well aligned with the implementation of co-benefits in the GAINS model.

*Line 21*: "lie in the in" -- typos here
Done. Fixed typo.

*Line 60-61*: the language here implies that it is both the volatility and reactivity that facilitate long-range atmospheric transport, whereas in actuality the reactivity decreases propensity to transport. This sentence could be edited to ensure accuracy.

Thank you for pointing this out. This introductory sentence is perhaps attempting to state too many facts at the same time and is thus insinuating a linear logic where there is none. However, we do not agree that all 'reactivity' decreases propensity to transport: Yes, Hg(0) may be oxidized to Hg(II), thus decreasing its propensity to undergo long-distance travel in the atmosphere. However, there are numerous documented instances where the opposite may also happen and oxidized Hg(II), either as an aqueous species or bound to a surface, may be reduced by environmental conditions and is thus volatilized and emitted into the atmosphere. This has been documented to take place at numerous contaminated sites (e.g. Brocza et al. 2019, DOI: 10.1021/acs.est.9b00619)

We have changed the text in Lines 62 ff. to clarify:

> Hg has a high (redox-)reactivity at ambient conditions, facilitating frequent species changes. Elemental mercury ($Hg^0$) exhibits high volatility and vapor pressure that are unique for a metal and lead to long-range atmospheric transport, subsequent deposition and re-emission of the metal and its derivative compounds, as well as methylation and subsequent bioaccumulation in the aquatic food chain as methyl mercury (Selin, 2009).

**Line 63-65:** *Long-range Hg damages were known before 2002 -- in particular, Hg was considered by CLRTAP in the mid-1980s. Again, here, rephrase to ensure accuracy.*

Done. Thank you for pointing this out. I have amended this sentence and cited accordingly:

*Lines 62 f.:*

> Mercury has been known to be highly toxic to humans since the Minamata Disease tragedy in the 1950s and increasing awareness of the global dimensions of the Hg problem led to its explicit inclusion in the Aarhus protocol on heavy metals in 1998, as part of the CLRTAP convention (Selin & Selin 2006, CLRTAP 1979).

**Line 74:** *the Minamata Convention defines releases as those to water and land, and emissions as those to air. Thus, this sentence is not fully correct -- I would recommend using these terms consistently with how the Convention defines them.*

Done. Re-phrased.

Lines 75 f.:

> "To break the cycle of Hg emissions to air, releases into the environment, and subsequent re-emissions and build-up of Hg pollution in the environment, the Minamata Convention on Mercury (MCM) has been adopted in 2013."

Lines 78 f.:

> "The MCM aims to "protect the human health and the environment from anthropogenic emissions and releases of mercury and mercury compounds" by targeting those emissions to air and releases to the environment different entry-points, such as trade, use in production, use in products, emission sources, and wastes. "

**Line 78-80:** *addressing releases does not require trade bans. The citation here to Giang et al. 2015 seems incorrectly placed.*

Addressed. We agree that the original phrasing allowed for multiple interpretations of the sentence and have re-phrased lines 81 ff.:

> "Mercury pollution is on one hand addressed by technical solutions, such as limiting emissions and releases through best available technology / best environmental practice (BAT/BEP) recommendations for Hg handling, industrial emissions or waste storage. On the other hand, there are provisions for regulatory action on other domains, such as severely limiting primary mercury mining and mercury trade, bans on specific products, and small scale or traditional (artisanal) gold mining practices, demonstrating a "life-cycle approach" to limiting Hg emissions (e.g. Selin 2014; UN 2013)."

**Line 100**: *it might be useful here to note how incredibly old the SRES scenarios are (this could be accomplished by citing them to their original source)*

Agreed and done. Added reference and comment:

Line 105 ff.:

> These emissions were projected to 2050 based on four climate scenarios from the IPCC Special Report on Emission Scenarios (SRES), published in 2000 (Nakicenovic et al., 2000).

***Lines 97-112:*** *beyond describing the existing scenarios it would be useful here to describe, at least qualitatively, what they project for emissions in terms of range*
Thank you for this comment, this is done. Emission projections have been added to lines 110-113. After careful consideration, we have decided against including this scenario in the discussion, as the base year, activity projections as well as some sector definitions are significantly different.

***Line 165:*** *it would be useful to describe the iPOG tool here for the unfamiliar reader (or at least expand the acronym)*
Done. Long name of tool inserted in Line 175.

***Line 213-217****: these are all very optimistic climate scenarios in terms of energy use. How does this compare with the current estimates of CO2 emissions from the Paris Agreement?*
According to the Climate Action Tracker, current real-world action is putting us on a track to 2.7°C warming by 2100 relative to the pre-industrial average. This is slightly more CO2-intensive than the World Energy Outlook's 2022 STEPS scenario (used as BAS in this manuscript). Climate tracker's most optimistic scenario, taking into account 'full implementation of all **announced** targets including net zero targets, LTSs and NDCs' projects a median global warming of 1.8°C by 2100. This would be similar to the WEO 2022 "Announced Pledges" scenario (CLIM1 in this study).

Reference:
The CAT Thermometer. December 2023. Available at: https://climateactiontracker.org/global/cat-thermometer/. Copyright © 2023 by Climate Analytics and NewClimate Institute.
IEA 2022. World Energy Outlook 2022. Paris.

***Line 237:*** *this sentence implies that VCM phase-outs are mandated by the Minamata Convention, but that is incorrect.*
Done. Changed wording in Lines 251 ff.:
> The MCM mandates that the Hg intensity of VCM production needs to be reduced by 50% in 2020, relative to production in 2010. For this study, 2015 VCM production values were assumed to be constant based on the data reported in the GMA'18, but the Hg emission intensity was adjusted as mandated by the Minamata convention."

***Line 424-425****: it's interesting that the Hg reductions occur at a lower rate than CO2. Is this because of other sources? It's hard from the graph to determine the extent to which there is a proportional association for combustion sources with CO2 decreases? If not other sources, what is the intuition behind this? Perhaps there could be a plot of the ratio with time (maybe in the SI?)*
Thank you for this question.
I will focus my answer on the sectors COMB_POWER, COMB_IND and COMB_OTHER, the combustion sectors. There are several reasons why the relationship between Hg emissions and CO2 emissions may not be linear when looking at the aggregated sectors as a whole.
1. All combustion sectors consist of a multitude of sub-sectors, which have different CO2 and Hg emission intensities per PJ of fuel. Thus, depending on which sectors are seeing the biggest CO2 reduction, it might result in different Hg emissions.
2. Different fuels again have different Hg emission intensities. The COMB_ sectors include the combustion of different fossil fuels (5 different coal grades, gas, heavy fuel oil, gasoline, diesel, natural gas...), as well as biomass and renewables. The trends for these fuels differ in the energy scenarios (see e.g. Figure 3): e.g. biomass use is projected to increase in all three climate scenarios.
3. Mitigating CO2 emissions through end-of-pipe pollution control (CCS) is not very common, wheras air pollution control devices for PM, SO2, NOx and even Hg are widespread and deliver substantial co-benefits. Furthermore, clean air policies are rapidly evolving in some parts of the world (e.g. China) and have already been incorporated into GAINS as current

legislation. This means that Hg emissions are de-coupling more from CO2 emissions in the power sector and the global, average Hg emissions per PJ of e.g. coal combusted is lower in 2050 than it was in 2015. The same might hold true for CO2, but to a much smaller extent.

**Line 509-512**: *it would be interesting to compare this estimate quantitatively with previous estimates of emissions not covered by the Minamata Convention -- in the 2017 ICMGP synthesis in Ambio, the policy paper estimated this quantity at 97 Mg. This seems comparable, but maybe this estimate is a bit bigger? Is there a key sector missing in the previous study, or is this just a function of a different base year emissions inventory?*

Thank you for this suggestion! The Ambio 2017 estimate is using the 2013 Global Mercury Assessment as its basis, while GAINS is calibrated to its update form 2018. The following lines have been added, comparing GMA'18 and GAINS estimates:

L. 518 ff.:

> In this study, only roughly 10% (of the emissions computed by GAINS fall into the 'NOT_COVERED' category: 234.6 t Hg in 2015, increasing slightly to 259.8t by 2050 in BAS_CLE. The number includes emissions from residential and domestic coal combustion, biomass combustion, transport emissions, emissions related to oil refining, production of iron and steel, paper, fertilizer, glass and aluminium. This number is higher than the GMA'18 estimate (171.5 t Hg, not including paper, fertilizer and glass).

**Public access**: *please check to make sure the code is accessible upon official publication. Currently, the link refers to a password protected page that is only accessible to collaborators.*

Due to the licensing of some underlying data, it is not possible for us to give full access to the presented scenario without registration. It is possible to request access to the Global GAINS model for a reviewer – even while remaining anonymous, as the request would go through the model administrator, not via the authors (http://gains.iiasa.ac.at/models/gains_models4.html).

While it would not be possible to open all access to the model code, the GAINS methodology has been well documented (please refer to http://gains.iiasa.ac.at/models/gains_resources.html for a list of publications and technical reports). A publication detailing the mercury calculation is currently in the works.

That said, while we are not at liberty to disclose detailed exogenic inputs into the model, we are able to provide more detailed modelling results at a more detailed level than presented, e.g. Hg emissions per sector and per each of the 182 GAINS regions, upon request.

We have added made the following changes:
- Changed link to http://gains.iiasa.ac.at/models/gains_models4.html

**Response to Reviewer #2 (Francesco De Simone)**

**Comment:** *This study represents a milestone in the literature. The mercury community, both from the scientific and policy sides, waited for a similar paper for more than a decade. Of course, other similar studies were published, although only at a limited regional or sectoral scope. There is plenty of very useful information for any researcher in the area. On the flip side of the coin, it is difficult to thoroughly review all of the technical aspects covered by the paper. However, the methodology and the tool used have been very consolidated through the years and do not require a detailed review in this regard. Therefore, I will focus on the aspects I have personally faced in the recent past in my research activities.*

**Reply 1**: Thank you for for this kind feedback! We are in the process of also preparing and publishing a methodology report that contains detailed information on the computation of Hg control

strategies, in particular. Until then, we hope that the supplementary information, combined with existing GAINS reports (e.g. here http://gains.iiasa.ac.at/models/gains_resources.html) on other pollutants, will provide enough information. As mentioned to Reviewer #1, it is also possible to request more detailed Hg emission datasets (e.g. on a country level) or reviewer access to GAINS global.

*In this regard, the only critical issue I can see is not methodological and does not affect the quality or validity of the paper, but in my opinion requires it to be addressed by the authors before publication. It is strictly linked to ASGM, their Hg emission estimates, and, in particular, the associated uncertainty. The aspects of uncertainty associated with ASGM Hg emissions are covered through the text in many sections; however, this is not the case for the associated implications. The most critical factor is that, considering the relative weight of ASGM Hg emissions, the differences between the scenarios designed and depicted in Figure 3 (upper panel), Figure 4 (regional panel where ASGM apply), and Figure 5 are likely to be completely covered by the associated uncertainty (of ASGM Hg emissions). At first sight, this makes de facto indistinguishable most of the scenarios (except for 12_CLIM2_HgMFR, which considers the utopian ban of ASGM impossible for many reasons). One would argue that this makes the study somewhat inconclusive. Conversely, I believe this is a very important outcome to be considered by policymakers.*

**Reply 2:** This is done. We fully agree with this sentiment and appreciate the nudge to include it more explicitly in the discussion! Discussion has been added to the following lines:

L. 326 ff.: …

**African, Central and South American as well as Southeast Asian** Hg emissions are dominated by ASGM and the uncertainty intrinsic to these estimates (-36.8% up to +44.5%, see Table S12), eclipses trends in all other sectors such as small increases in cement and waste sectors. For all three regions, it is important to note that ASGM estimates for 2015 are subject to large uncertainty (Keane et al., 2023) and the purpose of projections in all scenarios can only be to show the influence of Hg policy such as the Minamata convention, as the activity is kept constant.

*As a reader, I would expect error bars to be present in the indicated figures; however, as a researcher, I believe I understand their absence. I could suggest authors realize a version of the figures (at least figure 5) that excludes ASGM, allowing a detailed evaluation of the scenarios for the other sectors.*

**Reply 3:** This is done. Thank you for this comment. We agree with the sentiment have thought long about how to adequately represent uncertainty of the results. Ultimately, we have decided to discuss uncertainty in the text, but display error bars only in Fig. S2 in the SI.
We have changed Fig. 5 to excluded ASGM and added the following discussion to the text (L. 434):

As ASGM emissions are not affected by climate policy in our model, they have been excluded from Fig. 5 for easier comparison of the remaining sectors.

[Figure]

Revised Fig. 5.

*ASGM is a complex social phenomenon rather than an industrial trend; linking it to large-scale gold makes sense for many aspects and countries but can be irrelevant for others. Of course, the authors have to make a choice, and this does not affect the quality of the paper or its publication. However, I suggest the authors underscore all these aspects in the pertinent sections and in the conclusion and expand par. 4 of Section 5.7 (line 547).*

**Reply 4:** This is done.

Added to Section 5.7 (4), L. 561:

> Similarly, policy and technical measures for ASGM emission reduction might be refined to better reflect the complexity of this social phenomenon.

Re-arranged Section 6 (Conclusion) to first address ASGM and expanded discussion, L. 571 ff.:

> Of all studied sectors, ASGM offers the largest absolute Hg reduction potential, but also the largest uncertainty in the emission estimates, and in world regions with significant ASGM contribution, all other emission sources fall within the uncertainty of ASGM emissions, making the studied scenarios (with exception of Hg-MFR) virtually undistinguishable. While caution should be exercised when interpreting absolute emission values, valuable information can be gained from considering trends across years and between scenarios, and sectors.

Additional precision in wording, L. 581f. :

> In all studied sectors except ASGM, emission increases can be dampened or reversed by conventional air pollution control measures.

Inclusion of ASGM in closing statement: L. 592 f.:

> Overall, the findings emphasize the importance of the ASGM sector for total Hg emission reduction. Further, implementing targeted Hg control policies in addition to stringent climate and air pollution abatement policies is vital achieve significant reductions in Hg emissions.

---

## Author Response (AR2)

**Response to Editor – Corrections**

**May 7, 2024**

**Manuscript EGUSPHERE-2024-41**

**Title :** "Global scenarios of anthropogenic mercury emissions"

**Corresponding Author:** Flora Maria Brocza, brocza@iiasa.ac.at

Dear Professor Zhang,

I am delighted to hand over the final version of our manuscript for publication.

Thank you for handling the review process so swiftly!

Please find replies and explanations of the final technical corrections below.

Kind regards,

Flora Brocza on behalf of the author team.

**General changes to uploaded manuscript**

- Table 3: changed from pdf to word table. The pdf table included slightly outdated numbers for OTHER and COMB_POWER among others. These were updated to be consistent with all graphs and also with the deposited data.

- Figure 4: the alignment between sub-plots was improved.

**Technical corrections: Response to Reviewer #1**

The revisions have improved the manuscript and its presentation.

Overall, I understand the revised justification for including only ambitious climate policy, and this is acceptable for publication. I'm not sure I quite agree with the author's take-away that they are able to robustly test whether "Hg emissions from combustion become relatively less important compared to other emission sources, and dwindle when compared to ASGM emissions" -- as they state in the response to reviewer -- since they don't include high-combustion-intensive scenarios. However, the text in the paper itself is appropriate.

Thank you for this response. It is duly noted that 'high combustion' scenarios would be very worthwhile additional scenarios. We hope that we can analyze a wider set of scenarios with region-specific policy relevance and perhaps more targeted policy scenarios in the future!

In a few places, the revisions introduced a few more inaccuracies which are outlined here. These should be fixed before final publication:

**Line 68:** Mercury has been known to be highly toxic to humans since the Minamata Disease tragedy in the 1950s --> this is incorrect, I suggest editing to "methyl mercury", or a rephrase, as Hg has been known to be highly toxic to humans for centuries (see https://direct.mit.edu/books/oa-monograph/4968/chapter/1901501/Human-Health-Mercury-s-Caduceus for a historical summary, including references).
**Reply:** Done. Thank you for this comment and also for the very useful reference! I had not read this chapter of the Selin&Selin book yet. I sincerely hope that my new phrasing does not introduce more inaccuracies!
L. 65 ff:
> "Mercury has been known to be highly toxic to humans for centuries, but global attention on its health impacts and in particular of methylmercury toxicity has been heightened since the Minamata Disease tragedy in the 1950s (e.g. Selin & Selin, 2020). "

**Line 70**: referring to "global dimensions" of the problem in relation to the Aarhus convention is slightly awkward, as this was officially not accounted for in this regional convention -- potentially "long-range transport" dimensions would be better?
**Reply:** Done. Thank you for this very useful suggestion!
L. 67:
> "Increasing awareness of the long-range transport of $Hg^0$ and the resulting cross-boundary nature of the Hg problem…"

**Line 82**: 147 ratifications is actually incorrect, as some parties have acceded, not ratified. Please change to 147 parties.
**Reply:** Done. Apologies – I have clearly mislabeled the number of (acceded) parties here. I have now referred back to the "Parties and Signatories" page of the Minamata Convention and reviewed the status of different countries. There are only 103 ratifications in the strict sense, so I have followed the terminology on the convention homepage and now refer to '128 signatories', as indicated on the top of the page.
L. 78, 79:
> "It entered into force in 2017 (UNEP, 2013) and there are 128 signatories to the Convention as of May 2024."

**Line 91:** the new text now refers to "severely limiting primary mercury mining" which has introduced an error, the Convention actually bans primary mercury mining!
**Reply:** Done. This might have been inference from my mother language, where I read "primary mercury mining and mercury trade" as one phrase. This is now specified:
L. 86, 87:
> "…., there are provisions for regulatory action on other domains, such as banning primary mercury mining and severely limiting mercury trade,… "

**Table 1**: Why was gasoline deleted here?
**Reply:** Gasoline was deleted as it is part of the 'Liquid Fuels' aggregation category. Upon revision, it was decided that it would be inconsistent to single out gasoline while aggregating all other liquid fuel categories.